# Body Contacts and Social Interactions in Captive Odontocetes Are Influenced by the Context: An Implication for Welfare Assessment

**DOI:** 10.3390/ani10060924

**Published:** 2020-05-26

**Authors:** Agathe Serres, Yujiang Hao, Ding Wang

**Affiliations:** 1Institute of Hydrobiology, Chinese Academy of Sciences, Wuhan 430050, China; wangd@ihb.ac.cn; 2Institute of Hydrobiology, University of Chinese Academy of Sciences, Beijing 100101, China

**Keywords:** agonistic interaction, bottlenose dolphin, emotional state, finless porpoise, pectoral fin contact, socio-sexual interaction

## Abstract

**Simple Summary:**

Even though species differ in terms of personality traits and responses to external stimuli, welfare-oriented studies conducted on odontocetes are mostly focused on bottlenose dolphins (*Tursiops truncatus*). Odontocetes are highly social animals; social behaviors are therefore interesting to investigate in relation to welfare. Video recording was conducted over a year on three groups of captive odontocetes, to record the frequency of social behaviors across different contexts. Captive odontocetes’ social behaviors, such as pectoral contacts, other body contacts, agonistic interactions or social play were influenced by the context and the patterns observed in this study suggest their potential usefulness to assess welfare in these animals, and that species and groups might react differently to a stimulus.

**Abstract:**

Research on the welfare of captive odontocetes has increased in recent years, but has been mostly focused on bottlenose dolphins (*Tursiops truncatus*). Few studies investigated potential welfare indicators using quantitative data linked to a range of conditions or stimuli that are thought to impact the animals’ emotional state. Since odontocetes are social animals that engage in various social interactions, these interactions might inform us on their welfare state. We investigated pectoral contact laterality and the effect of the context on several social behaviors in three groups of captive odontocetes (Yangtze finless porpoises, YFPs: *Neophocaena asiaeorientalis asiaeorientalis*; East-Asian finless porpoises, EAFPs: *N. a. sunameri*, and bottlenose dolphins, BDs). Animals exhibited patterns depending on the time of the day for most of the social behaviors we analyzed; social separation was associated with lower rates of social behaviors for the two analyzed groups (YFPs and BDs), the accessibility to several pools was associated with higher rates of social behaviors for BDs. The effect of enrichment, disturbances and public presence was less clear and strongly depended on the group, the type of enrichment and disturbance. Our results confirm that captive odontocetes’ social behaviors are influenced by the context, and that, depending on the group, some of them, such as pectoral contacts, other body contacts, agonistic interactions or social play exhibit consistent patterns across contexts. Monitoring these behaviors might be useful to adapt the captive management to each species and group. The different responses among the three studied groups confirm that species and groups react differently to a stimulus and therefore, management decisions should be species/group specific. We recommend that more studies should be conducted to validate our findings in other groups of odontocetes under human care.

## 1. Introduction

Research on the welfare of zoo animals has expanded in recent years [1]. Various definitions of welfare exist. The standard definition of animal welfare usually includes the five needs, stating that animals should (1) be housed in a suitable environment; (2) be fed a suitable diet; (3) be free to express normal behaviors; (4) be housed with or apart from other animals; and (5) be protected from any pain, suffering, disease or injury [2]. However, nowadays, a “feelings-based” definition of welfare is often adopted [3,4,5]. Welfare can be defined as a balance between positive and negative emotional states and positive experiences being linked with good welfare, whereas aversive experiences are linked with poor welfare [6]. Studies and reviews about captive odontocetes’ welfare pointed out the lack of reliable work on parameters that could be used to assess welfare: welfare indicators [7,8]. Such indicators can be behavioral, physiological, or cognitive parameters (e.g., cognitive bias [9,10]). In order to find behaviors that could be good candidates, effects of different conditions or stimuli on the occurrence of behaviors and interactions should be studied. For instance, there is a need to investigate how husbandry and/or management decisions impact captive odontocetes’ social life, including the social separation of individuals [11]. 

It is assumed that behavioral data-often easier to collect than physiological, health or cognitive data-could be as informative as other categories of potential welfare indicators [12,13]. Social behaviors were suggested to be of particular importance in odontocete species that often exhibit a high degree of sociality [14,15], and might be linked with the animals’ welfare state, and observing their variations might inform us about this state. Social animals, such as odontocetes, benefit, but also suffer from, this high level of sociality. On one side, they are very likely to suffer from social stress [11], but on the other side, the bonds they form with their conspecifics might help them to overcome stressful situations [10]. 

Among social behaviors, social interactions, such as agonistic interactions and social play, were suggested to be potential indicators of welfare in bottlenose dolphins [16,17]. Social play is thought to reflect positive welfare because it has been described as occurring only when an individual’s primary needs are fulfilled [18,19], but the fact that play can evolve into agonistic interactions implies the need to use this parameter with caution [20]. Because pectoral contacts have been suggested to reflect social bonds [21,22] and to be negatively correlated with the occurrence of aggressive behaviors [23], they should be investigated as a potential measure of positive welfare. Body contacts, such as pectoral contacts between individuals, have been frequently studied and described as affiliative behaviors, which can be used to assess welfare [24,25,26,27]. Body contacts previously described in odontocetes include simple contact (contact with no movement), rubbing (contact with movement) and petting (pectoral fin contact with movement, [28]). Such behaviors have been described in wild Indo-Pacific bottlenose dolphins, *Tursiops aduncus*: [29,30,31,32], spinner dolphins, *Stenella longirostris*: [33], Atlantic spotted dolphins, *Stenella frontalis*: [28,34], belugas, *Delphinapterus leucas*: [35], rough-tooth dolphins, *Steno bredanensis*: [36], sperm whales, *Physeter macrorhynchus*: [37] and captive animals bottlenose dolphins, *Tursiops truncatus*: [38,39], spinner dolphins: [33], Commerson’s dolphins: *Cephalorhynchus commersoni*: [40], Yangtze finless porpoises, *Neophocaena asiaeroientalis asiaeorientalis*: [41]. The behavioral context in which such body contacts occur has been recently investigated but not published [42], but the modulation by the environmental context has never been studied in odontocetes. Socio-sexual interactions (i.e., interactions involving at least two individuals engaging in sexual behaviors, such as sexual contacts, sexual rubbing or mounting) are frequent in odontocetes and might have a role in bond establishment, especially for males [43,44,45,46]. Such interactions require a close proximity, and are probably influenced by the social status of individuals [47,48]. In addition, an abnormally high frequency of sexual activity has been suggested to be a potential sign of boredom in captive animals [49], making it an interesting behaviour to study in relation to welfare.

Compared to other odontocetes species, we know much about bottlenose dolphins’ social life and most welfare studies have been conducted on this species [50], leaving other less represented species poorly studied. Such species might have different social lives and might not exhibit the same responses or react the same as bottlenose dolphins do in captive settings [51]. Therefore, studying different species’ responses to various contexts is useful to better understand their behavior and to adapt the captive management to each species and even each group. Finless porpoises (FPs)–including the freshwater species, the Yangtze finless porpoise (*Neophocaena asiaeorientalis asiaeorientalis:* YFP) and the East Asian finless porpoise (*Neophocaena asiaeorientalis sunameri:* EAFP)–are classified as endangered (critically endangered for the YFP) species under the criteria of the IUCN red list of threatened species [52]. In addition to in situ conservation, captive groups of FPs are kept in several Asian aquaria or research centers. However, even though captive breeding is an ultimate goal for these species, their welfare in captivity was never studied. Investigating welfare indicators for these animals might aid the management of social groups and help to increase breeding success. Since bottlenose dolphins (*Tursiops truncatus*: BDs) are the most studied odontocete species [53], investigating the same parameters in a BD group and in FP groups might be useful to compare our findings with the available literature. We analyzed the effect of environmental and social factors on the social behavior (pectoral fin contacts, other body contacts, agonistic interactions, socio-sexual interactions and social play) in two captive groups of FPs and one of BDs. Investigating variation patterns of such behaviors depending on the context will provide preliminary information on the potential link between these behaviors and situations that are thought to be positive or negative for the animals. Finding consistent patterns between behaviors and across contexts will help to determine which behaviors are interesting to investigate further, in relation with welfare in each species.

## 2. Material and Methods

### 2.1. Subjects, Housing and Group Composition

Data were collected from early September 2017 to late October 2018. The study focused on five YFPs observed in Baiji Dolphinarium, Institute of Hydrobiology, Chinese Academy of Sciences, four EAFPs and five BDs observed in Haichang Polar Ocean world, both located in Wuhan, China (Table 1). All YFPs were initially housed in a 20 m length, 7 m width and 3.5 m depth kidney-shaped pool, connected to a 10 m diameter and 3.5 m depth round pool. When the group was separated, a gate allowing animals to see each other was closed between these two pools. An unconnected 13 m diameter and 3.2 m depth round pool was used starting February 2017 to house the female F7 and the male Taotao until F7 gave birth. Taotao was transferred back in the two-pool complex just before the birth, and after birth, F7 stayed alone in this pool with her calf. For group management reasons (i.e., management of pregnant females), the social grouping changed several times during the data collection period. Since the three females gave birth during summer 2018, three calves were also present during certain periods of the data collection (two of them died quickly and therefore were only present for less than two weeks after their birth, and the third one was present from its birth until the end of the data collection). EAFPs were always kept together in a 13.75 m length, 8 m width, 5.8 m depth rectangular pool. BDs were kept in a three-pool complex, including two 8.86 m diameter, 5 m deep wide round pools (“small pools”) connected to the main pool of 27.44 m long, 12 m wide, and 6 m deep (“large pool”), where public presentations occurred. Depending on the observation sessions, animals had access to one or two pools. A new female arrived in the facility on January 16th 2018, and the other female was placed with her, starting from January 23rd 2018. When males and females were separated, females were kept in one of the round pools and males in the other round pool and/or in the main pool. On two occasions, the social grouping changed for a couples of days (a female with two males and another female with one male). The female Beila was absent from several morning observations because of a medical treatment administrated in the medical pool over the course of one month.

YFPs were subject to four to six training sessions a day during which were fed between 3 and 3.5 kg of thawed and/or live fish in total. Occasional visitors were allowed to watch animals, both from the surface and from underwater windows. EAFPs were not trained but had three feeding sessions a day, with a total fed of between 2.5 and 3 kg of thawed fish per day, sometimes including live fish. BDs participated in three training sessions and two public presentations a day, within which they were fed between 10 and 13 kg of thawed fish. During holidays, public presentations replaced trainings (up to five public presentations a day).

Environmental enrichment was provided in each group. Enrichment is the addition of sensory stimuli and/or choices in the environment [54], in order to increase behavioral opportunities for animals. Goals of environmental enrichment are (1) to increase behavioral diversity; (2) to reduce the frequency of abnormal behaviors; (3) to increase the range of normal behavior patterns; (4) to increase positive utilization of the environment; (5) to increase the ability to cope with challenges in a more normal way [55]. Here, animals were provided human-made objects (i.e., toys) or live fish (for YFPs and EAFPs) at times decided by caretakers, and caretakers frequently interacted with BDs and YFPs outside of training sessions. In addition, all pools were frequently cleaned by divers and/or caretakers, scrubbing the upper part of the pools’ walls. 

### 2.2. Data Collection

Previous to the data collection, a one-month ad libitum preliminary study was conducted to identify and familiarize with the study subjects and to build a common ethogram for the three species (Table 2).

The data collection lasted 14 months for YFPs, and 12 months for EAFPs and BDs. For each species, data were collected with no gap (no week without data collection during the data collection period). Each group was monitored two days a week, at least three times a day (in early morning, at noon and in the early afternoon). Recording sessions always occurred between feedings/training sessions/public presentations. Each session lasted 15 min and consisted of video and voice recording, using up to six cameras to monitor each group depending on the pool configuration. When monitoring the kidney-shaped pool of YFPs, two underwater and two overhead monitoring cameras were used. We used one underwater camera for the connected round pool and two underwater and one overhead cameras for the disconnected round pool. EAFPs who were always housed in the same pool were monitored using two Xiaoyi 4K cameras (Yi Technology, Shanghai, China) placed in front of two underwater windows. The monitoring of BDs was achieved using two Xiaoyi 4K cameras in front of a bubble-shaped window situated five meters deep in the main pool and three other Xiaoyi 4K cameras placed on a bridge above the water to monitor the surface of the main pool and the round pools. Because the observation bridge was located between the main pool and the two round pools and because the two round pools were not very deep, the behavioral recording was good enough from the surface for these two pools. For each group, the cameras covered around 90% of every pool with quality, enabling visual analysis of the videos. Synchronously with the video recording, the observer was always narrating what animals were doing, using a voice recorder or the cameras’ audio recording to ensure the identification of each individual and to ease the video analysis.

During every data collection day and for each observation session, environmental parameters were recorded. These parameters included social housing (all animals together: “not separated”, group divided in subgroups: “separated”), pools in which animals were observed (only for BDs, since they were not housed in the same pool depending on days and time of the day), presence of visitors (“none”, “few” or “many”), presence and type of enrichment, and every unusual event that occurred (“disturbance”, Table 3). In the “enrichment” category, we decided to include all stimuli provided by caretakers in order to stimulate animals, and in the “disturbance” category, we included stimuli that were not meant to be enriching, but that had to occur in facilities.

### 2.3. Analysis

Videos were visually analyzed to record all occurrences of previously defined social behaviors and interactions categories for each individual using incident sampling ([56], Table 2). This analysis consisted of watching recorded videos to report the frequency of selected behaviors on a sheet of paper before entering it into an excel table. Videos were paused and replayed as many times as needed to ensure a reliable coding. To take the group size into account in the analysis, the frequency of each behavior was weighted by the number of animals present in the group for each recording session.

Statistical analysis was performed using R 3.5.2. The side preference for pectoral contacts and the effect of environmental parameters on the frequency of each behavioral category (right and left pectoral fin contacts, other body contacts, agonistic interactions, socio-sexual interactions, social play) were analyzed for each species using generalized linear mixed effect models (GLMMs) for Poisson distributed data (“glmer ()” function from lme4 package, [57]). For testing pectoral side preference, the number of contacts was included in models as response variable and the side as predictor. For testing the effect of environment, the frequency of the tested behavior was included in models, as the response variable and all the environmental parameters were included together as predictors. Because of sample balance issues and because we did not find any worthy and interpretable interaction between factors, interaction was included in models. For all models, the individual ID and the date were included as random factors. Each model was run once for each behavior and for each species (e.g., one model to test agonistic interactions’ frequency as response variable for YFPs). Overdispersion was checked for each model and a case-level random factor was added if needed (overdispersion ratio > 2). Collinearity was tested via the variance inflation factor (VIF), with no major issue (no VIF > 10, [58]). For each model, a model selection was conducted by comparing models’ Akaike information criterion (AIC): the model with the lowest AIC was selected. Wald chi square tests were used to extract p values from models. Pairwise comparisons between levels of each variable were run using the same models with an appropriate sub-setting. 

## 3. Results

Data were collected over the course of 142 days for YFPs, 100 days for EAFPs and 100 days for BDs. In total, every YFP individual was monitored 135 h (540 15 min recording sessions), 76 h for EAFPs (304 15 min recording sessions), and 80 h for BDs (320 15 min recording sessions).

### 3.1. Pectoral Fin Contact Laterality

YFPs used their right pectoral fin significantly more than their left pectoral fin to touch other individuals (χ² = 5.94, df = 1, *p* = 0.014). EAFPs and BDs did not exhibit a significant preference for a pectoral fin (respectively: χ² = 0.51, df = 1, *p* = 0.48; χ² = 1.98, df = 1, *p* = 0.16). 

### 3.2. Effect of Environmental and Social Parameters

#### 3.2.1. Time of the Day

YFPs engaged in left and right pectoral contact more often in the morning than at noon (Table 4). EAFPs’ left pectoral contacts were significantly less frequent in the morning than at noon and in the afternoon, and their right pectoral contacts were significantly less frequent in the morning than at noon and in the afternoon. BDs did not exhibit any significant pattern depending on the time of the day for left or right pectoral contacts. The frequency of other body contacts did not vary depending on the time of the day for YFPs. The frequency of other body contacts was significantly higher in the morning than at and in the afternoon, and higher in the afternoon than at noon for EAFPs. BDs engaged in other body contacts significantly more frequently in the morning than at noon. 

The frequency of agonistic interactions was significantly higher at noon than in the morning and in the afternoon, and tended to be higher in the morning than in the afternoon for YFPs. The frequency of agonistic interactions was significantly higher in the afternoon than at noon for EAFPs. BDs engaged in agonistic interactions significantly more frequently in the morning than at noon and in the afternoon, and more frequently at noon than in the afternoon.

The frequency of socio-sexual behavior was significantly higher at noon than in the morning and in the afternoon for YFPs. The frequency of socio-sexual interactions did not significantly vary depending on the time of the day for EAFPs. BDs engaged in socio-sexual interactions significantly more frequently in the morning than in the afternoon. BDs exhibited no significant pattern depending on the time of the day for social play.

#### 3.2.2. Separation

YFPs and BDs engaged in right and left pectoral contact significantly less when separated than when altogether (Table 5). The frequency of other body contacts did not vary depending on the social grouping for YFPs, and BDs engaged in other body contacts significantly less when separated than when not. Agonistic interactions and socio sexual interactions were significantly more frequent when YFPs and BDs were not separated than when they were separated. BDs engaged in social play significantly less when separated than when not separated. 

#### 3.2.3. Housing Pool

For BDs, left pectoral contacts were significantly more frequent when BDs were housed in the large pool than when in the small pool or when having access to both pools, and right pectoral contacts tended to be less frequent when having access to both pools than when housed in the small Table 6). Other body contacts were significantly more frequent when BDs had access to both pools than when housed in the small or the large pool. Agonistic interactions were significantly more frequent when BDs had access to both pools than when housed in the small or the large pool. Socio-sexual interactions were not significantly impacted by the housing pool. Social play was significantly more frequent when BDs had access to both pools than when housed in the small or the large pool.

#### 3.2.4. Enrichment

For YFPs, the frequency of right pectoral contact was significantly lower when live fish was provided, or when human(s) were present with toy(s), than when no enrichment was provided (Table 7). For EAFPs, the frequency of left pectoral contact was significantly higher when human(s) were present than when no enrichment was, and the frequency of right pectoral contact was significantly higher when human(s) were present than when no enrichment was provided. For BDs, the frequency of left pectoral contact was significantly lower when human(s) were present than when no enrichment was provided. The frequency of other body contacts did not vary depending on the presence of enrichment for YFPs. The frequency of other body contacts was significantly higher when human(s) or toy(s) were present than when no enrichment was for EAFPs. The frequency of other body contacts was significantly lower when human(s) were present and higher when toy(s) were provided, than when no enrichment was provided for BDs. Agonistic interactions were significantly more frequent for YFPs when toy(s) or human(s) were present, than when no enrichment was. The frequency of agonistic interactions was significantly higher when toy(s) were provided, but lower when new objects were, than when no enrichment was, for EAFPs. The frequency of agonistic interactions was significantly lower when human(s) were present with toy(s), than when no enrichment was provided for BDs. For YFPs, socio-sexual interactions were significantly less frequent when live fish was provided than when no enrichment was. Socio-sexual interactions were significantly more frequent when toy(s), human(s) or new objects were present than when no enrichment was, for EAFPs. For BDs, the frequency of socio-sexual interactions was significantly lower when human(s) were present than when no enrichment was provided. The frequency of social play was significantly lower for BDs when human(s) were present and higher when toy(s) were provided, than when no enrichment was provided.

#### 3.2.5. Disturbances

For YFPs, right and left pectoral contacts were significantly less frequent during pool cleaning and social events and more frequent during noisy events and other disturbances, than when no disturbance occurred (Table 8). For EAFPs, left pectoral contacts were significantly more frequent during disturbances than when no disturbance occurred. For BDs, left and right pectoral contacts were significantly more frequent during pool cleaning and social events than when no disturbance occurred. For YFPs, other body contacts’ frequency was significantly lower during pool cleaning, noisy events, social events, or other disturbances than when no disturbance occurred. For EAFPs, other body contacts’ frequencies were not significantly impacted by the occurrence of disturbances. For BDs, other body contacts were significantly more frequent during pool cleaning and social events than when no disturbance occurred. Agonistic interactions frequency did not vary depending on the presence of disturbances for YFPs or EAFPs. The frequency of these interactions was significantly lower during noisy events than when no disturbance occurred for BDs. Socio-sexual interactions were significantly more frequent when noisy events or other disturbances occurred than when no disturbance did for YFPs. For EAFPs, socio-sexual interactions were less frequent when pool cleaning or other disturbances occurred than when no disturbance did. The frequency of these interactions was significantly lower during pool cleaning, noisy events or other disturbances than when no disturbance occurred for BDs. For this species, the frequency of social play was significantly higher during pool cleaning and social events than when no disturbance occurred.

#### 3.2.6. Public

For YFPs, left pectoral contacts were significantly less frequent when many visitors were present than when few of them or none of them were (Table 9). For EAFPsboth side pectoral contacts frequencies were significantly lower when many visitors were present than when few of them or none of them were. The presence of visitors did not significantly impact the frequency of left or right pectoral contacts in BDs. The frequency of other body contacts did not vary depending on the presence of visitors for YFPs and BDs. The frequency of other body contacts was significantly higher when many visitors were present than when none were. For YFPs and BDs, agonistic interactions frequency did not vary depending on the presence of visitors. For EAFPs, agonistic interactions frequency was significantly higher when many visitors were present than when none were. Socio-sexual interactions frequency did not vary depending on the presence of visitors for YFPs, EAFPs and BDs. The frequency of social play was significantly lower when few visitors were present than when none were for BDs.

## 4. Discussion

### 4.1. Pectoral Fin Contact Laterality

YFPs used their right pectoral fin significantly more than their left pectoral fin, which might originate from a hemispheric bias in these animals [59]. These YFPs have been shown to mostly swim clockwise [60], which places their left pectoral fin on the wall side. This position may explain why YFPs used their right pectoral fin more often. EAFPs and BDs did not exhibit a significant preference for a pectoral fin. A study on 27 BDs that analyzed pectoral fin contact bias suggested an ambidextrousness [61], which is congruent with what we found for EAFPs and BDs. This absence of bias could also be the result of different biases among individuals in the group, since we did not analyze each animal’s side preference. Behavioral lateralization has been suggested to be of potential importance in the evaluation of an animal’s welfare state. Since the left hemisphere controls the behavior performed in non-stressful situations, whereas the right hemisphere deals with stressful emotions, stressed animals are thought to use the right hemisphere more and to exhibit a left behavioral bias [62,63]. This study area is expanding, but remains poorly studied in odontocete species. Our results suggest that such laterality bias should be further investigated and that their link with the context should be studied. A right pectoral fin bias in YFPs could be indicative of a better emotional state for the individuals of this group than for the two other groups that did not exhibit any bias. However, since laterality has been shown to vary depending on individuals [64], an analysis on an individual level and depending on the context is needed to validate this hypothesis.

### 4.2. Effect of Environmental and Social Parameters

#### 4.2.1. Time of the Day

YFPs engaged in pectoral contacts more often in the morning, while EAFPs displayed it more in the afternoon. For other body contacts, EAFPs and BDs displayed it the most in the morning. YFPs and BDs engaged in agonistic and socio-sexual interactions more often in the morning and at noon, while EAFPs engaged in agonistic interactions the most in the afternoon. Most social activities of four BDs (*Tursiops aduncus*) were shown to be globally more frequent in the early afternoon [65]. However, it has been shown that captive odontocetes were less active in the afternoon [17,66]. A higher rate of social interactions in the morning and at noon for YFPs and BDs, as well as the higher rate of pectoral fin contacts and body contacts that are often displayed during social interactions such as swimming in contact, socio-sexual interactions or social play, could be due to this higher activity level. Differences between groups might result from differences between facilities (e.g., routine schedule). It might also reflect the different social function of body contacts, agonistic and socio-sexual interactions among groups for instance. In addition, each of the three studied species does not live in the same habitat in the wild, and they are therefore subject to different environments, resources distribution, presence of other species and anthropogenic pressure, for instance. Such factors may have influenced the animals’ behaviour and their daily patterns [67,68,69,70,71,72]. The pattern depending on the time of the days we found here can be useful for management issues in each facility. Routines, as well as unusual events, can be discussed and adapted to the animals’ activity rhythm in order not to disturb their social life, or to improve their environment’s stimulating value. Enrichment can be provided in the afternoon, when animals interact less with their conspecifics, for instance.

#### 4.2.2. Separation

Social separation might impact the animals’ relationships and the way they interact, both in a positive or negative way [11]. Here, social separation was associated with lower rates of social play for BDs. This lower level of playfulness when separated has already been shown in these groups when analyzing solitary play [73]. Separation has also been shown to increase circular swimming and affiliative behaviors, such as social swimming (i.e., synchronous swimming, contact swimming, group swimming), in the presently studied groups [60]. Such social condition could be a stressful situation, eliciting higher cohesion between animals to cope with stress, resulting in lower rates of agonistic interactions. Here, for YFPs, except once (for the separation of pregnant females), separation did not follow any particular event or did not take place in a situation that required such decision, but was usually done as a way to increase the animals’ attention during training, and to separate males from females outside of the breeding period. Social separation might sometimes be needed when social issues arise [11] and can improve the situation for some individuals. However, such a decision should be carefully discussed and its impacts should be observed and analyzed, to ensure it has the impact that was expected, or at least that it does not worsen the situation.

#### 4.2.3. Housing Pool

Since the small pool was much smaller than the large one, body contact could have occurred more often when housed in it than when having access to a larger space, because of the animals’ obvious proximity. Body contact in odontocetes has been suggested to mostly serve social and hygienic functions, and to be mostly affiliative [24,39,68]. Such contacts, including pectoral fin contacts, also occur as socio-sexual behaviors [38,74], and are potential analogs of grooming behavior that terrestrial mammals engage in [75]. The fact that animals engaged in pectoral and other body contacts more often when housed in a larger pool could reflect a higher rate of affiliative interactions when having access to a larger space, since these body contacts were often displayed during such interactions. A lower body contact rate might indicate a lower rate of affiliative interactions. Since affiliative behaviors are crucial in the development and maintenance of bonds in odontocetes [14,15,76], the space animals have access to could therefore be influencing the relationship between individuals and the way they bond. Regarding social interactions, we suggest that, unlike agonistic interactions and social play that were intense and involved fast swimming and chasing, socio-sexual interactions rarely involved much swimming behavior and thus did not require a large space. This difference could explain why agonistic interactions and social play were more frequent when having access to a larger space, while socio-sexual interactions were not impacted by the housing pool. This result is congruent with the higher rate of fast swimming in this group that was found when they had access to both pools [60]. Housing environment, including pool configuration, has been shown to influence captive belugas’ behavior with more dynamic and variable swim patterns when having access to multiple pools [77]. In addition, when giving polar bears (*Ursus maritimus*) the opportunity of accessing their indoor enclosures, they engaged more often in social play [78]. Such configuration provides opportunities for animals to make choices, such as withdrawing from other individuals or choosing where to swim or to rest. We suggest that such minor modifications to the environment and management, including access to different spaces can increase welfare, and that they should be tested further in captive odontocetes. Another factor that has to be considered when analyzing the housing pools’ influence on the behaviour of dolphins is the habitat that they live in in the wild. For instance, it has been shown that BDs that usually live in shallow areas in the wild would spend less time in large pools than in smaller ones [76]. Here, the exact provenance of the animals was unknown, and therefore, we cannot link the animals’ behaviour in each pool with the environment they live in in the wild.

#### 4.2.4. Enrichment

Body contacts were less frequent for YFPs and BDs, but more frequent for EAFPs when enrichment was present. EAFPs engaged in pectoral contacts more often when humans or toys were provided, which could have been caused by the close proximity between animals when they were interacting with them: we noticed that EAFPs often observed others playing with toys or interacting with humans, and were often staying close (or even in contact) to them when doing so. EAFPs were also coming to underwater windows in pairs to interact with people, staying very close to each other, and therefore frequently touching each other (one individual interacting and one observing or both individuals interacting). Conversely, YFPs and BDs interacted with toys or humans alone. Agonistic interactions were more frequent in YFPs and BDs, but less frequent in BDs when enrichment was present. For YFPs, toys and humans were always provided in low numbers (1–2 humans/toys), resulting in conflicts in accessing this limited resource. For EAFPs, the presence of a new object, unlike the presence of toy(s), was associated with a lower frequency of agonistic interactions. Animals might have reacted to novelty by increasing group cohesion, therefore decreasing aggressive behaviors. For BDs, the presence of humans together with toys was associated with a lower rate of agonistic behaviors between animals, which is an interesting result to further investigate. It has already been shown that BDs are likely to interact with caretakers outside of training sessions and interact with toys more frequently when caretakers are involved [79,80]; combining these two kinds of enrichment might therefore increase their enriching properties. EAFPs engaged in socio-sexual interactions more often when toys were provided. This pattern could be linked to the males’ sexual behaviors (sexual rubbing, mounting) towards conspecifics that were interacting with a toy. Finally, the fact that BDs engaged in social play less when humans were present, but more when toys were, reflects the way in which they interacted with each enrichment type: they often played socially with objects (e.g., two individuals pulling two sides of a rope), while they did not play socially with humans. Play has often been suggested to be indicative of positive affective states in animals [81,82,83,84,85], because it is not present when animals are experiencing unfavorable environmental conditions (e.g., food shortage) or negative states (e.g., pain [86,87,88]). This rewarding, pleasurable and exciting activity still has to be used cautiously to assess welfare [18], because depending on the context, the affective states animals experience while engaging in play may not always be fully positive [89]. Authors highlighted the need to identify species-specific contexts in which an increase in play could be considered an indicator of good welfare and encouraged work on the contexts where different kinds of play behaviors appear [20,90]. Providing animals an opportunity to choose which type of enrichment to interact with is a way for animals to control their environment and is often beneficial for captive individuals [78]. When aiming to give more opportunities of control to captive animals, the needs of a given species and even of each individual must be considered when utilizing different types of enrichment, social groupings, or habitat configurations [91,92,93]. Differences of reaction to each type of enrichment among the three groups we studied might therefore reflect preferences for certain type of enrichment [79,80], and also the way they interact with each type of enrichment (alone, together, with or without distance between individuals). The differences we found between groups could be linked with species-specific traits, such as shyness for YFPs, and individual differences might play a role in the reaction to enrichment [79]. In addition, the animals’ management and especially the training routine and the relationship with caretakers might have been a factor that influenced the preference for a certain type of enrichment and even explain the preference for interacting with caretakers rather than socializing with conspecifics. Here, EAFPs were not trained, and had much less contacts with humans than YFPs and BDs, and BDs were much more stimulated than YFPs (i.e., higher diversity of behaviors asked during training, higher diversity of enrichment, enrichment provided more often, caretakers interacting with them more often outside of training sessions). Such conditions probably influenced the results that we found here.

#### 4.2.5. Disturbances

Body contacts were lower for YFPs but higher for EAFPs and BDs when disturbances occurred. However, for YFPs, depending on the type of disturbance, pectoral contacts were either more (noise, other disturbances) or less (pool cleaning, social events) frequent. The fact that odontocetes have been observed to stay closer when facing disturbances (killer whales: [94], bottlenose dolphin: [95]) may be an explanation for the increase of body contacts for EAFPs and BDs when disturbances occurred. Conversely, the lower frequency of body contacts frequency observed in YFPs might reflect another strategy to face the disturbances we analyzed. Differences in reaction between groups for the different types of perturbation might be linked with group-specific sensitivity and behavioral response, as well as facility differences in terms of frequency of disturbances’ occurrence for instance [51]. Since odontocetes are sensitive to noise [96,97], the fact BDs engaged significantly less in agonistic interactions when noisy events occurred might reflect a vigilant state, preventing animals to interact. This is congruent with the fact that socio-sexual interactions also decreased in this context and with all other disturbance types. EAFPs also engaged less frequently in socio-sexual interactions when disturbances occurred. It has already been shown that the analyzed disturbances were linked with a higher circular swimming distance and a higher rate of fast swimming in the studied groups [60]. The decrease of agonistic and socio-sexual interactions rates might therefore be due to a matter of time budget (e.g., animals were engaged in other behaviors). Socio-sexual interactions are thought to play a role in bond establishment, especially for males [43,44,45,46]. A long-term decrease in such interactions could therefore indicate social issues in a group. It is currently hard to explain why socio-sexual behaviors increased in potentially stressful contexts for YFPs, but since they only increased during disturbance types that did not elicit the most obvious response (i.e., pool cleaning resulted in obvious changes in swimming behaviors [60]), it might indicate that the level of disturbance could play a role in the animals’ response. Finally, social play was more frequent in BDs during disturbances (pool cleaning and social events). The opposite was found in a previous study [17], but in this work, only the noise of construction work was included as a disturbance. For BDs, unlike noisy events, pool cleaning and social events were quite frequent (every day for pool cleaning), and might have elicited an excitement state, resulting in more play behaviors.

#### 4.2.6. Public

Pectoral contacts were less frequent in YFPs and EAFPs, but other body contacts were more frequent in EAFPs when many visitors were present. In addition, agonistic interactions were more frequent in EAFPs when many visitors were present. The presence of the public has been shown to affect animals’ behavior in captivity [98,99,100,101]. For gorillas (*Gorilla gorilla gorilla*), negative behavioral indicators (e.g., aggression, stereotypies, auto-grooming) increased with noise and visitors’ density (e.g., [99]). A study conducted on several zoo species found an increased vigilance and activity when public noise increased [102]. Since visitors are often a source of noise, the auditory acuity and sensitivity to noise of an animal impacts its responses to their presence [103]. Odontocetes that are sensitive to noise [96,97] might be strongly impacted by such a presence. Here, the presence of many visitors might have elicited a decrease in affiliative behaviors, that could explain the lower rate of body contact. For EAFPs, the higher rate of body contact and agonistic interactions when many visitors were present could have been caused by a kind of excitement/frustration state, because the presence of visitors is a parameter that animals cannot control [104]. No study directly investigated the impact of visitors on captive odontocetes. However, a study on siamangs (*Hylobates syndactylus*) and white-cheeked gibbons (*Hylobates leucogenys*) found that they did not exhibit a difference in the time spent engaged in social behaviors when many or few visitors were present, but spent more time hiding from the public when many visitors were present [105]. Such responses highlight the need for animals to have visual barriers in their enclosures in order to enable them to withdraw [105]. However, these withdraw possibility issues have never been investigated in odontocetes and might not be applicable. In addition, public presence is not always negative; it can be neutral or even positive: visitors can represent a kind of enrichment [104,106,107]. Such enriching value could have elicited the animals’ curiosity and led to lower body contacts, and even to increased agonistic behaviors to appropriate the visitors’ attention. More research is needed on this topic.

During the data collection, YFPs and EAFPs were never observed engaging in social play as described in other odontocete species, which is why we did not analyze this behaviour for these two species. Since object play and locomotor play were observed in both porpoise species, it does not seem to be a matter of playfulness. The absence of social play could be a species-specific characteristic in the way animals interact with each other. For instance, YFPs and EAFPs engaged in socio-sexual behaviours more often than BDs. Such differences in the interactions that animals engage in might reflect a different way to bond.

This study’s findings are only true for the three investigated groups, and further research should be conducted to observe if similar patterns can be found in other groups. In addition, some behavioral categories such as pectoral fin contacts should be investigated deeper with detailed categories of contacts (e.g., while swimming, while resting) and with the response of the receiver analyzed. For agonistic interactions, socio-sexual interactions and social play that vary a lot in terms of intensity and form, the intensity and the displayed behaviors should also be taken into account and be used to categorize bouts. The influence of the context on the pectoral fin contact side should also be investigated further, to see if some contexts elicit a higher use of the right or left pectoral fin.

## 5. Conclusions

This study highlighted a side bias for pectoral contacts in YFPs, and left and right pectoral contacts were not always influenced by the context following the same pattern, suggesting that further investigation on behavioral laterality is needed. Animals exhibited patterns depending on the time of the day for most of the social behaviors analyzed, with a higher activity level in the morning for two groups (YFPs and BDs) and in the afternoon for the third one (EAFPs). Such patterns have to be taken into account for setting routine management schedules (e.g., enrichment) and particular events, for instance. Social separation was associated with lower rates of social interactions, including socio-sexual interactions, agonistic interactions, and social play. We suggest that separation might affect the animals’ social lives and relationships, and that their behavior should be monitored before and after each separation event to observe positive or adverse effects of such a management decision. Conversely, the accessibility to several pools was associated with higher rates of social behaviors (except socio-sexual interactions) for BDs. However, this result might be specific to this facility where the housing pool was extremely small, and the animals were extremely energetic. The effect of enrichment and disturbances was less clear: body contact was less frequent for YFPs and BDs, but more frequent for EAFPs when enrichment was present. Agonistic interactions were more frequent in YFPs and EAFPs, but less frequent in BDs, when enrichment was present. Competition over enrichment (e.g., toys) might have been different because of the amount of enrichment provided at the same time (often lower for FPs than for BDs). We suggest that enrichment should be provided in a large amount (i.e., many items at the same time), allowing all animals to access it. For BDs, unlike the presence of toys or humans alone, the presence of humans together with toys seemed to reduce aggressive behaviors between animals, which is an interesting result to further investigate. Combining different enrichment might increase their enrichment properties. We suggest that humans might have been more attractive and elicited certain responses from BDs that were not observed for the other two species. Both species-specific traits and management routine probably influenced the results we found. Body contacts were lower for YFPs but higher for EAFPs and BDs when disturbances occurred, potentially indicating different strategies in facing stressful situations. BDs engaged significantly less in agonistic and socio-sexual interactions when disturbances occurred, which might reflect a vigilant state, preventing animals to interact. These results show that captive odontocetes’ social behaviors are influenced by the environment and the patterns we found suggest that some of them could be useful to assess welfare in these animals. For instance, the patterns we found for social play in BDs are in line with previous results, indicating that it could indicate positive emotional states. Pectoral contacts, other body contacts and agonistic interactions’ frequency globally increased for the three studied groups with stimuli that are thought to be positive for animals (enrichment, access to a larger space) and decreased in conditions that could be stressful or elicit negative emotional states (disturbances, separation). However, the differences of responses among the three studied groups confirms that species and groups react differently to a stimulus, and each group should be subject to a preliminary monitoring to determine basal frequencies and patterns, depending on the time of the day of each behavior. In addition, an increased frequency of agonistic interactions could be a sign of social problems in groups (e.g., social dominance issues, inappropriate social grouping) [9,49], therefore, this parameter has to be investigated further and used with caution. More research on other groups of odontocetes should be conducted, in order to confirm whether or not these behaviors or interactions could be useful tools to assess welfare. The behaviors and interactions we suggested to be potential welfare indicators should be used together with other parameters (other behaviors, physiology, health, cognition), to obtain an accurate welfare assessment [9].

## Figures and Tables

**Table 1 animals-10-00924-t001:** Catalogue of studied individuals’ features (YFP: Yangtze finless porpoise, EAFP: East-Asian finless porpoise, BD: bottlenose dolphin).

Name	Species	Sex	Age (year)	Weight (Kg)	Length (m/h)	Facility
**Duoduo**	YFP	M	8	NA	157	Baiji dolphinarium, IHB
**F7 ***	YFP	F	8	NA	145	Baiji dolphinarium, IHB
**F9 ***	YFP	F	8	NA	145	Baiji dolphinarium, IHB
**Taotao**	YFP	M	14	NA	156	Baiji dolphinarium, IHB
**Yangyang ***	YFP	F	11	NA	147	Baiji dolphinarium, IHB
**Xiaomeng**	EAFP	F	4	33	1.43	Haichang Wuhan Polar Ocean park
**Xiaomi**	EAFP	M	7	31	1.60	Haichang Wuhan Polar Ocean park
**Xiaoxi**	EAFP	M	4	41.5	1.49	Haichang Wuhan Polar Ocean park
**Xiaozhuang**	EAFP	M	7	48	1.70	Haichang Wuhan Polar Ocean park
**Ailun**	BD	M	13	280	2.74	Haichang Wuhan Polar Ocean park
**Beila**	BD	F	11	250	2.52	Haichang Wuhan Polar Ocean park
**Jiesen**	BD	M	14	290	2.69	Haichang Wuhan Polar Ocean park
**Luoke**	BD	M	13	260	2.70	Haichang Wuhan Polar Ocean park
**R ***	BD	F	15	260	2.55	Haichang Wuhan Polar Ocean park

* pregnant females

**Table 2 animals-10-00924-t002:** Catalogue of social behaviors and interactions used for the video analysis.

Behavioral Category	Behavior and Description	
**Social Body Contacts**		
	Pectoral fin contact	Any contact between the pectoral fin of an individual and another individual’s body (genital parts excluded). This behavior is only recorded for the initiator (the pectoral fin contact giver)
	Right pectoral fin contact	Pectoral contact using the right pectoral fin
	Left pectoral fin contact	Pectoral contact using the left pectoral fin
	Other body contact	Any body contact that is not involving pectoral fins (sexual contacts and aggressive contacts excluded)
**Social Interactions**		
	Agonistic	Any interaction including at least one aggressive behavior such as chasing, tail slapping, threatening, biting etc. with a context allowing to distinguish it from social play (for BDs)
	Socio-sexual	Any interaction including sexual behaviors such as genital looking, sexual contact, swimming with genitals in contact with another’s body, or mounting
	Social play (only for BDs)	Any interaction including at least one behavior that was previously described as occurring during play such as chasing, tail slapping, biting etc. with a context allowing to distinguish it from agonistic interactions

**Table 3 animals-10-00924-t003:** Environmental and social factors’ features (adapted from Serres et al. 2019).

Species	Separation	Housing Pool	Disturbance	Enrichment	Visitors
YFP	Not separated	NA	None	None	None
	Separated (gate between groups allowing visual and acoustic contact)		Noisy event (construction work noise or loud people noise)	Toy(s) (balls)	Few (<5 persons in front of underwater windows or next to the pool: employees or visitors)
			Pool Cleaning (diver and/or caretaker scrubbing from the surface using long handle brushes)	Interaction with human(s) (caretakers interacting with animals outside of training sessions at the surface of from underwater windows)	Many (>5 persons in front of underwater windows or next to the pool: visitors)
			Social event (right after separation or reunion of groups, after the birth or the death of a calf)	Human(s) + toy(s)	
				Live fish	
			Other (shoal of small fish in the pool, water level unusually high or low)	New object in the water (Soundtrap, stretcher, experiments’ material, new toy)	
EAFP	NA	NA	None	None	None
			Noisy event (construction work, microphone speakers)	Toy(s) (ball, Soundtrap)	Few (<15 visitors in front of underwater windows)
				Human(s) (public interacting through underwater windows)	Many (>15 visitors in front of underwater windows)
			Pool cleaning (diver or small boat)	New object (Soundtrap, filtration items)	
			Other (water unusually high, unknown person sampling water)		
BD	Not separated	Small (housing pool)	None	None	None
	Separated (gate allowing visual and acoustic contact)	Large (public presentation pool)	Noisy event (construction work or public presentation rehearsal)	Toy(s) (balls, buoys, ropes + buoys)	Few (<8 persons next to the pool, usually employees)
		Large + small	Pool cleaning (divers)	Human(s) (caretakers interacting with animals outside of training sessions)	
			Social event (separation attempt or arrival of a new individual)	Human(s) + toy(s)	
			Other (water sampling, underwater sound recording)		

**Table 4 animals-10-00924-t004:** Fitted means and standard errors of the frequency of each social behaviour depending on time of the day for Yangtze finless porpoises (YFPs), East-Asian finless porpoises (EAFPs) and bottlenose dolphins (BDs). Outputs come from Wald chi-squared tests run on Generalized linear mixed models, significant outputs are bolded. For each species and each behaviour, items sharing the same letter did not differ significantly, items that do not share the same letter differ significantly (post-hoc tests with sequential Bonferroni correction).

Species		YFPs			EAFPs			BDs		
Time of the Day		Morning	Noon	Afternoon	Morning	Noon	Afternoon	Morning	Noon	Afternoon
**Left Pectoral Contact**	**Mean**	1.18	1.14	1.13	1.97	3.64	3.02	1.01	1.01	1.01
	**SE**	3.46	2.72	2.47	0.34	0.64	0.55	1.21	1.18	1.02
	**Output**	**χ² = 11.11, df = 2, *p* = 0.004**			**χ² = 57.33, df = 2, *p* < 0.001**			χ² = 0.71, df = 2, *p* = 0.701		
	**Pairwise**	a	b	b	b	a	a			
**Right Pectoral Contact**	**Mean**	1.26	1.21	1.18	2.25	3.25	2.86	1.02	1.02	1.02
	**SE**	0.17	0.14	0.12	0.47	0.67	0.60	0.01	0.01	0.01
	**Output**	**χ² = 14.16, df = 2, *p* < 0.001**			**χ² = 21.65, df = 2, *p* < 0.001**			χ² = 0.72, df = 2, *p* = 0.698		
	**Pairwise**	a	b	b	c	b	a			
**Other Body Contact**	**Mean**	1.01	1.01	1.01	9.98	5.49	4.86	1.02	1.01	1.01
	**SE**	0.26	0.19	0.26	0.38	0.28	0.26	1.23	0.67	0.41
	**Output**	χ² = 0.59, df = 2, *p* = 0.742			**χ² = 12.98, df = 2, *p* = 0.002**			**χ² = 4.73, df = 2, *p* = 0.094**		
	**Pairwise**				a	c	b	a	b	ab
**Agonistic**	**Mean**	1.11	1.14	1.10	4.06	3.26	4.36	1.42	1.28	1.21
	**SE**	3.27	4.37	3.06	0.22	0.18	0.23	0.16	0.11	0.09
	**Output**	**χ² = 6.36, df = 2, *p* = 0.042**			**χ² = 10.24, df = 2, *p* = 0.006**			**χ² = 32.62, df = 2, *p* < 0.001**		
	**Pairwise**	b	a	b	ab	b	a	a	b	c
**Socio-Sexual**	**Mean**	1.09	1.14	1.08	7.01	6.68	6.50	1.76	1.60	1.48
	**SE**	9.43	14.00	7.79	2.03	1.92	1.88	0.36	0.30	0.25
	**Output**	**χ² = 11.30, df = 2, *p* = 0.004**			χ² = 0.67, df = 2, *p* = 0.71			**χ² = 8.25, df = 2, *p* = 0.016**		
	**Pairwise**	b	a	b				a	ab	b
**Social Play**	**Mean**	NA			NA			1.35	1.28	1.32
	**SE**							0.11	0.09	0.10
	**Output**							χ² = 2.80, df = 2, *p* = 0.247		
	**Pairwise**									

**Table 5 animals-10-00924-t005:** Fitted means and standard errors of the frequency of each social behaviour, depending on social grouping for Yangtze finless porpoises (YFPs) and bottlenose dolphins (BDs). Outputs come from Wald chi-squared tests run on Generalized linear mixed models, significant outputs are bolded. For each species and each behaviour, items sharing the same letter did not differ significantly, items that do not share the same letter differ significantly (post-hoc tests with sequential Bonferroni correction).

Species		YFPs		BDs	
Social Grouping		Altogether	Separated	Altogether	Separated
**Left Pectoral Contact**	**Mean**	1.16	1.15	1.02	1.01
	**SE**	3.16	2.93	2.29	1.00
	**Output**	**χ² = 8.79, df = 1, *p* = 0.012**		**χ² = 19.25, df = 1, *p* < 0.001**	
	**Pairwise**	a	b	a	b
**Right Pectoral Contact**	**Mean**	1.29	1.20	1.05	1.01
	**SE**	0.19	0.13	0.02	0.00
	**Output**	**χ² = 28.26, df = 1, *p* < 0.001**		**χ² = 11.68, df = 1, *p* < 0.001**	
	**Pairwise**	a	b	a	b
**Other Body Contact**	**Mean**	1.01	1.01	1.33	1.01
	**SE**	0.34	0.20	2.33	0.40
	**Output**	χ² = 2.04, df = 1, *p* = 0.154		**χ² = 63.94, df = 1, *p* < 0.001**	
	**Pairwise**			a	b
**Agonistic**	**Mean**	1.12	1.11	1.47	1.27
	**SE**	3.65	3.33	0.18	0.11
	**Output**	**χ² = 11.50, df = 1, *p* = 0.003**		**χ² = 19.76, df = 1, *p* < 0.001**	
	**Pairwise**	a	b	a	b
**Socio-Sexual**	**Mean**	1.21	1.07	1.97	1.56
	**SE**	19.93	7.29	0.43	0.28
	**Output**	**χ² = 49.92, df = 1, *p* < 0.001**		**χ² = 8.28, df = 1, *p* = 0.004**	
	**Pairwise**	a	b	a	b
**Social Play**	**Mean**	NA		2.58	1.24
	**SE**			0.34	0.08
	**Output**			χ² = 230.07, df = 1, *p* < 0.001	
	**Pairwise**			a	b

**Table 6 animals-10-00924-t006:** Fitted means and standard errors of the frequency of each social behaviour depending on housing pool for bottlenose dolphins (BDs). Outputs come from Wald chi-squared tests run on Generalized linear mixed models, significant outputs are bolded. For each species and each behaviour, items sharing the same letter did not differ significantly, items that do not share the same letter differ significantly (post-hoc tests with sequential Bonferroni correction).

Species		BDs		
Pool		Small	Large	Both
**Left Pectoral Contact**	**Mean**	1.01	1.02	1.01
	**SE**	1.07	1.68	0.68
	**Output**	**χ² = 10.33, df = 2, *p* = 0.006**		
	**Pairwise**	b	a	b
**Right Pectoral Contact**	**Mean**	1.02	1.03	1.01
	**SE**	0.01	0.01	0.01
	**Output**	**χ² = 4.90, df = 2, *p* = 0.086**		
	**Pairwise**	a	a	b
**Other Body Contact**	**Mean**	1.01	1.01	1.02
	**SE**	0.70	0.62	1.29
	**Output**	**χ² = 58.11, df = 2, *p* < 0.001**		
	**Pairwise**	b	b	a
**Agonistic**	**Mean**	1.25	1.31	1.54
	**SE**	0.11	0.13	0.20
	**Output**	**χ² = 37.78, df = 2, *p* < 0.001**		
	**Pairwise**	b	b	a
**Socio-Sexual**	**Mean**	1.64	1.53	1.70
	**SE**	0.31	0.27	0.34
	**Output**	χ² = 2.48, df = 2, *p* = 0.288		
	**Pairwise**			
**Social Play**	**Mean**	1.28	1.26	1.82
	**SE**	0.09	0.08	0.22
	**Output**	**χ² = 95.30, df = 2, *p* < 0.001**		
	**Pairwise**	b	b	a

**Table 7 animals-10-00924-t007:** Fitted means and standard errors of the frequency of each social behaviour depending on enrichment for Yangtze finless porpoises (YFPs), East-Asian finless porpoises (EAFPs) and bottlenose dolphins (BDs). Outputs come from Wald chi-squared tests run on Generalized linear mixed models, significant outputs are bolded. For each species and each behaviour, items sharing the same letter did not differ significantly, items that do not share the same letter differ significantly (post-hoc tests with sequential Bonferroni correction).

Species		YFPs					EAFPs				BDs			
Enrichment		None	Toy(s)	Human(s)	Human(s) and Toy(s)	Fish	None	Toy(s)	Human(s)	New Object	None	Toy(s)	Human(s)	Human(s) and Toy(s)
**Left Pectoral Contact**	**Mean**	1.15	1.14	1.14	1.06	1.10	2.11	2.54	5.84	7.70	1.01	1.02	1.00	1.01
	**SE**	3.00	2.77	2.67	1.14	2.06	0.37	0.46	0.88	1.16	1.28	1.76	0.26	0.66
	**Output**	**χ² = 6.16, df = 4, *p* = 0.192**					**χ² = 116.95, df = 3, *p* < 0.001**				**χ² = 12.27, df = 3, *p* = 0.007**			
	**Pairwise**						b	ab	a	ab	a	ab	b	ab
**Right Pectoral Contact**	**Mean**	1.23	1.18	1.21	1.06	1.14	1.99	2.73	5.86	2.91	1.02	1.01	1.03	1.01
	**SE**	0.15	0.12	0.14	0.06	0.10	0.40	0.57	1.01	0.67	0.01	0.01	0.01	0.01
	**Output**	χ² = 11.78, df = 4, *p* = 0.019					**χ² = 109.24, df = 3, *p* < 0.001**				χ² = 4.46, df = 3, *p* = 0.216			
	**Pairwise**	a	ab	ab	b	b	b	ab	a	ab				
**Other Body Contact**	**Mean**	1.01	1.01	1.01	1.00	1.07	7.57	5.38	17.32	2.85	1.01	1.01	1.00	1.01
	**SE**	0.22	0.31	0.30	0.01	2.66	0.34	0.26	0.50	0.40	0.78	0.94	0.27	0.75
	**Output**	**χ² = 9.08, df = 4, *p* = 0.591**					**χ² = 30.51, df = 3, *p* < 0.001**				**χ² = 5.01, df = 3, *p* = 0.025**			
	**Pairwise**						b	a	a	ab	b	a	c	abc
**Agonistic**	**Mean**	1.11	1.14	1.17	1.20	1.10	3.05	4.08	4.27	2.78	1.31	1.28	1.42	1.15
	**SE**	3.40	4.30	5.07	5.97	3.04	0.18	0.21	0.24	0.33	0.13	0.12	0.17	0.07
	**Output**	**χ² = 5.16, df = 4, *p* = 0.023**					**χ² = 10.02, df = 3, *p* = 0.018**				**χ² = 7.53, df = 3, *p* = 0.006**			
	**Pairwise**	b	a	a	ab	ab	b	a	abc	c	a	ab	ab	b
**Socio-Sexual**	**Mean**	1.10	1.08	1.20	1.07	1.06	5.32	6.78	7.15	27.99	1.67	1.59	1.38	1.40
	**SE**	10.13	7.75	19.27	7.52	6.45	1.55	1.94	2.10	11.55	0.32	0.30	0.21	0.22
	**Output**	**χ² = 6.51, df = 4, *p* = 0.011**					**χ² = 38.51, df = 3, *p* < 0.001**				**χ² = 20.51, df = 3, *p* < 0.001**			
	**Pairwise**	a	ab	ab	ab	b	b	a	a	a	a	ab	b	ab
**Social Play**	**Mean**	NA					NA				1.31	1.47	1.18	1.32
	**SE**										0.10	0.14	0.07	0.11
	**Output**										**χ² = 4.42, df = 3, *p* = 0.035**			
	**Pairwise**										b	a	c	abc

**Table 8 animals-10-00924-t008:** Fitted means and standard errors of the frequency of each social behaviour depending on disturbances for Yangtze finless porpoises (YFPs), East-Asian finless porpoises (EAFPs) and bottlenose dolphins (BDs). Outputs come from Wald chi-squared tests run on Generalized linear mixed models, significant outputs are bolded. For each species and each behaviour, items sharing the same letter did not differ significantly, items that do not share the same letter differ significantly (post-hoc tests with sequential Bonferroni correction).

Species		YFPs					EAFPs				BDs				
Disturbance		None	Noisy Event	Pool Cleaning	Social Event	Other	None	Noise	Pool Cleaning	Other	None	Noisy Event	Pool Cleaning	Social Event	Other
**Left Pectoral Contact**	**Mean**	1.16	1.26	1.03	1.05	1.42	2.74	1.73	3.41	2.05	1.01	1.00	1.04	1.05	1.00
	**SE**	3.16	4.92	0.69	0.94	7.31	0.50	0.29	0.61	0.40	1.54	0.37	4.45	5.67	0.00
	**Output**	**χ² = 125.21, df = 4, *p* < 0.001**					**χ² = 17.78, df = 3, *p* < 0.001**				**χ² = 19.12, df = 4, *p* < 0.001**				
	**Pairwise**	b	a	c	c	a	b	a	a	a	b	ab	a	a	ab
**Right Pectoral Contact**	**Mean**	1.23	1.34	1.07	1.08	1.65	2.75	2.17	2.68	3.04	1.02	1.01	1.03	1.04	1.01
	**SE**	0.15	0.22	0.06	0.06	0.38	0.58	0.46	0.57	0.67	0.00	0.01	0.01	0.02	0.01
	**Output**	**χ² = 178.18, df = 4, *p* < 0.001**					χ² = 3.15, df = 3, *p* = 0.369				χ² = 4.46, df = 3, *p* = 0.216				
	**Pairwise**	b	c	a	a	c									
**Other Body Contact**	**Mean**	1.01	1.01	1.00	1.00	1.00	6.75	8.40	4.68	4.34	1.01	1.00	1.02	1.07	1.01
	**SE**	0.29	0.27	0.10	0.13	0.15	0.30	0.48	0.29	0.43	0.00	0.00	0.01	0.03	0.02
	**Output**	**χ² = 27.01, df = 4, *p* < 0.001**					χ² = 4.92, df = 1, *p* = 0.177				χ² = 9.44, df = 4, *p* = 0.051				
	**Pairwise**	a	b	b	b	b					b	abc	a	a	ab
**Agonistic**	**Mean**	1.11	1.11	1.12	1.12	1.15	3.78	3.57	3.97	3.92	1.32	1.05	1.24	1.34	1.23
	**SE**	3.38	3.27	3.77	3.77	4.43	0.20	0.26	0.23	0.35	0.13	0.03	0.11	0.14	0.11
	**Output**	χ² = 2.08, df = 4, *p* = 0.722					χ² = 0.28, df = 3, *p* = 0.963				**χ² = 12.18, df = 4, *p* = 0.004**				
	**Pairwise**										a	b	ab	ab	ab
**Socio-Sexual**	**Mean**	1.07	1.19	1.12	1.08	1.60	7.94	7.40	3.62	1.67	1.68	1.14	1.49	1.63	1.24
	**SE**	7.48	17.77	11.70	7.70	48.86	2.26	2.34	1.03	0.63	0.33	0.10	0.26	0.32	0.15
	**Output**	**χ² = 9.27, df = 4, *p* = 0.002**					**χ² = 92.22, df = 3, *p* < 0.001**				**χ² = 15.27, df = 4, *p* < 0.001**				
	**Pairwise**	b	a	ab	ab	a	a	ab	b	b	a	b	b	ab	b
**Social Play**	**Mean**	NA					NA				1.28	1.44	1.88	1.59	1.18
	**SE**										0.09	0.15	0.23	0.17	0.08
	**Output**										**χ² = 67.54, df = 4, *p* < 0.001**				
	**Pairwise**										b	ab	a	a	ab

**Table 9 animals-10-00924-t009:** Fitted means and standard errors of the frequency of each social behaviour, depending on visitors for Yangtze finless porpoises (YFPs), East-Asian finless porpoises (EAFPs) and bottlenose dolphins (BDs). Outputs come from Wald chi-squared tests run on Generalized linear mixed models, significant outputs are bolded. For each species and each behaviour, items sharing the same letter did not differ significantly, items that do not share the same letter differ significantly (post-hoc tests with sequential Bonferroni correction).

Species		YFPs			EAFPs			BDs	
Visitors		None	Few	Many	None	Few	Many	None	Few
**Left Pectoral Contact**	**Mean**	1.14	1.14	1.31	3.70	2.90	1.60	1.01	1.00
	**SE**	2.82	2.82	5.60	0.65	0.53	0.24	1.22	0.87
	**Output**	**χ² = 15.83, df = 2, *p* < 0.001**			**χ² = 62.53, df = 2, *p* < 0.001**			χ² = 2.15, df = 1, *p* = 0.142	
	**Pairwise**	a	b	a	a	a	b		
**Right Pectoral Contact**	**Mean**	1.22	1.20	1.20	3.17	2.94	1.80	1.02	1.02
	**SE**	0.15	0.14	0.14	0.66	0.62	0.34	0.01	0.01
	**Output**	χ² = 14.16, df = 2, *p* = 0.12			**χ² = 38.902, df = 2, *p* < 0.001**			χ² = 0.00, df = 1, *p* = 0.986	
	**Pairwise**				a	a	b		
**Other Body Contact**	**Mean**	1.01	1.01	1.00	9.03	6.20	4.57	1.01	1.01
	**SE**	0.24	0.25	0.06	0.38	0.29	0.27	0.76	0.64
	**Output**	χ² = 1.44, df = 2, *p* = 0.49			**χ² = 7.12, df = 2, *p* = 0.028**			χ² = 0.11, df = 1, *p* = 0.744	
	**Pairwise**				b	ab	a		
**Agonistic**	**Mean**	1.12	1.11	1.16	3.17	3.91	4.48	1.28	1.35
	**SE**	3.57	3.27	4.77	0.19	0.20	0.24	0.12	0.14
	**Output**	χ² = 1.39, df = 2, *p* = 0.498			**χ² = 6.65, df = 1, *p* = 0.010**			χ² = 3.11, df = 1, *p* = 0.776	
	**Pairwise**				b	ab	a		
**Socio-Sexual**	**Mean**	1.11	1.08	1.10	6.03	6.99	6.80	1.60	1.64
	**SE**	10.86	8.26	10.32	1.77	2.00	2.00	0.30	0.32
	**Output**	χ² = 2.14, df = 2, *p* = 0.343			χ² = 2.64, df = 2, *p* = 0.267			χ² = 0.11, df = 1, *p* = 0.737	
	**Pairwise**								
**Social Play**	**Mean**	NA			NA			1.38	1.15
	**SE**							0.12	0.05
	**Output**							**χ² = 52.95, df = 1, *p* < 0.001**	
	**Pairwise**							a	b

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
