# Peer review of "Body Contacts and Social Interactions in Captive Odontocetes Are Influenced by the Context: An Implication for Welfare Assessment"

_animals, 2020, doi:10.3390/ani10060924_

Round 1

Reviewer 1 Report

Much improved. Many thanks to the authors for incorporating that suggestions of the reviewers. One question remains, how are the authors controlling Experimenter-wise error with all of the analyses performed? I have added some comments to the pdf.

Author Response

Dear Reviewer,

Thank you for these new corrections. I followed all the suggestions you made in the PDF. About the experimenter-wise error, I forgot to mention that I applied a sequential Bonferroni correction. I added it on the tables' captions.

Thank you again

Reviewer 2 Report

Overall, this is an interesting study that provides novel findings to the cetacean welfare literature. Of particular note is the comparison of the same metrics between bottlenose dolphins and the less studied finless porpoises, as the demonstration of group- and species-specific responses to stimuli and contexts is extremely relevant for welfare management of captive groups and is generally missing from the existing literature. Some minor revisions would assist with the clarity of the manuscript, and a few major revisions to extend the findings or provide a more balanced discussion.  

Minor revisions:

  • Lines 43-45: use of five needs would be more modern and up to date, particularly as the UK source used is from 1979. Animal Welfare Act 2006 uses 5 needs so would be more appropriate.
  • Some small grammatical issues throughout, e.g. lengthy sentences. These mean readability could be improved overall
  • Line 110: rewording needed of ‘spited’, not clear what was intended here
  • Line 127: ‘…during which they were fed between…’
  • Line 166: how exactly were visitor numbers counted, i.e. in general how many were considered ‘few’ vs ‘many’?
  • Line 207: ‘depending on the time of day’ repeated unnecessarily
  • The results section is extremely long because of the amount of factors tested across groups. It would be helpful to present some of this in a summary table to help provide a bit more clarity and also to allow the reader to more easily make comparisons across species/groups. Although there are tables provided for each species separately, this still limits easy comparisons. I would recommend presenting time of day, social grouping, enrichment, disturbance, and visitors each as separate tables, with all three species included in each table. Delay to training and pool could be presented in its own table.
  • Discussion (lines 349-352): unclear why there is the repetition here of the hours of data collection, recommend removal
  • Discussion section 4.2.1, lines 382-385; it would be helpful here if an example was provided of how any particular element of a species-specific habitat, resource distribution etc is known to link to their activity pattern and whether this corresponds well with this study’s findings
  • Lines 417-425; would be worth considering here the link between space use and species-specific wild habitat choices, e.g. https://www.researchgate.net/publication/10907132_Effects_of_Pool_Size_on_Free-Choice_Selections_by_Atlantic_Bottlenosed_Dolphins_at_One_Zoo_Facility/figures?lo=1&utm_source=google&utm_medium=organic
  • Line 472: typo, ‘whether’ should presumably be ‘either’

Major revisions:

  • Individual ID was included as a random factor in models, but not discussed at all, even though it was referred to in discussion section 4.1 that analysis on an individual level could be used to validate the hypotheses that a right pectoral fin bias could be indicative of a better emotional state. Given this, it seems greatly short-sighted to not 1) present the individual dolphin results for each factor (perhaps as supplementary material) and 2) incorporate any known animal factors linked to individual behavioural measures in the discussion. There is a recent paper showing the individual variation in lateralised gaze duration (Lilley et al., 2020 – International Journal of Comparative Psychology) that may be useful for reference to this individual approach.
  • Line 186: why were no interactions between factors tested? Surely this could have a really important outcome. This needs a sound justification and/or to be included in analyses.
  • Table 2: why was social play only considered for BD? This seems like a potentially significant oversight for the other species (even if it didn’t occur at all, as they were housed socially this would be noteworthy)
  • Lines 412-413; this is a really interesting suggestion, and would benefit from additional exploration. It would also be further supported by considering literature on social bonding and links with these affiliative behaviours, e.g. Moreno et al., 2017 (https://psycnet.apa.org/record/2017-33838-001)
  • Lines 500-508: this is leaning towards biased, interpreting both the reduction and increase in body contacts in two different groups as negative. As well as evidence that visitors can have a negative impact, there is also evidence that for certain species/enclosure setups, they can have a neutral or even positive effect (Hosey, 2007). Should be considered here that it is possible for the presence of visitors to have produced a curiosity response resulting in decreased body contacts. Crucial point is that more information is needed, given that this is purely correlational there isn’t enough information to be able to interpret the behaviour changes one way or the other.

Author Response

Dear Reviewer, 

Thank you for your useful comments. I followed them, I hope the changes I made will suit you. In bold are my answers to your comments.

  • Lines 43-45: use of five needs would be more modern and up to date, particularly as the UK source used is from 1979. Animal Welfare Act 2006 uses 5 needs so would be more appropriate.

I changed it

  • Some small grammatical issues throughout, e.g. lengthy sentences. These mean readability could be improved overall

Corrected

  • Line 110: rewording needed of ‘spited’, not clear what was intended here

I reworded it

  • Line 127: ‘…during which they were fed between…’

Corrected

  • Line 166: how exactly were visitor numbers counted, i.e. in general how many were considered ‘few’ vs ‘many’?

Information in Table 3

  • Line 207: ‘depending on the time of day’ repeated unnecessarily

Corrected

  • The results section is extremely long because of the amount of factors tested across groups. It would be helpful to present some of this in a summary table to help provide a bit more clarity and also to allow the reader to more easily make comparisons across species/groups. Although there are tables provided for each species separately, this still limits easy comparisons. I would recommend presenting time of day, social grouping, enrichment, disturbance, and visitors each as separate tables, with all three species included in each table. Delay to training and pool could be presented in its own table.

We changed tables and added statistical outputs in to lighten the results part

  • Discussion (lines 349-352): unclear why there is the repetition here of the hours of data collection, recommend removal

This was a mistake

  • Discussion section 4.2.1, lines 382-385; it would be helpful here if an example was provided of how any particular element of a species-specific habitat, resource distribution etc is known to link to their activity pattern and whether this corresponds well with this study’s findings

I added litterature here

  • Lines 417-425; would be worth considering here the link between space use and species-specific wild habitat choices, e.g. https://www.researchgate.net/publication/10907132_Effects_of_Pool_Size_on_Free-Choice_Selections_by_Atlantic_Bottlenosed_Dolphins_at_One_Zoo_Facility/figures?lo=1&utm_source=google&utm_medium=organic

I added this

  • Line 472: typo, ‘whether’ should presumably be ‘either’

Corrected

Major revisions:

  • Individual ID was included as a random factor in models, but not discussed at all, even though it was referred to in discussion section 4.1 that analysis on an individual level could be used to validate the hypotheses that a right pectoral fin bias could be indicative of a better emotional state. Given this, it seems greatly short-sighted to not 1) present the individual dolphin results for each factor (perhaps as supplementary material) and 2) incorporate any known animal factors linked to individual behavioural measures in the discussion. There is a recent paper showing the individual variation in lateralised gaze duration (Lilley et al., 2020 – International Journal of Comparative Psychology) that may be useful for reference to this individual approach.

Since we did not go into individual analysis here, we prefer not to include this data and not to discuss individual differences too much in order not to confuse readers. We added this reference to justify our statement.

  • Line 186: why were no interactions between factors tested? Surely this could have a really important outcome. This needs a sound justification and/or to be included in analyses.

The information was added

  • Table 2: why was social play only considered for BD? This seems like a potentially significant oversight for the other species (even if it didn’t occur at all, as they were housed socially this would be noteworthy)

It did not occur at all in the two other groups, we added this in the discussion

  • Lines 412-413; this is a really interesting suggestion, and would benefit from additional exploration. It would also be further supported by considering literature on social bonding and links with these affiliative behaviours, e.g. Moreno et al., 2017 (https://psycnet.apa.org/record/2017-33838-001)

Thank you for the paper, I have been searching for such article for a long time

  • Lines 500-508: this is leaning towards biased, interpreting both the reduction and increase in body contacts in two different groups as negative. As well as evidence that visitors can have a negative impact, there is also evidence that for certain species/enclosure setups, they can have a neutral or even positive effect (Hosey, 2007). Should be considered here that it is possible for the presence of visitors to have produced a curiosity response resulting in decreased body contacts. Crucial point is that more information is needed, given that this is purely correlational there isn’t enough information to be able to interpret the behaviour changes one way or the other.

Right, we added discussion on this point

This manuscript is a resubmission of an earlier submission. The following is a list of the peer review reports and author responses from that submission.

Round 1

Reviewer 1 Report

Having read several other papers by the authors on similar topics, the authors did a good job of parsing out the data for this study to contribute additional information to the field. There is some redundancy between the measures reported in this paper and a couple of other publications, but I believe the data are new so acceptable. 

The abstract was helpful and readable and accurate.

The intro covered the majority of the available literature. The authors did a nice job covering most of the cetacean literature as well as pulling in relevant terrestrial literature. The use of theses and dissertations was present, but I will leave that issue to the editors of the journal. Line 75, I would recommend adding in some way - "recently investigated but not published [40], and the modulation..." to address one of those unpublished sources. The sentence on socio-sexual behavior in line 77 is widowed and needs a home and probably a little more discussion. It was the only behavior not expanded on in the intro. Otherwise, all of the major DVS (pec fin, other contact, play, social play, agonistic intx, affiliative swims, socio-sex) were discussed within the current literature. 

The method was described well except for how the coding of the videos was actually conducted. Perhaps I missed this information? Also, was reliability assessed? All incidents were recorded for the key behaviors, so I am assuming frequency of bouts of the various social interactions were the used measures? What about the pec fin contact coding. . .I was a little surprised by the results in some cases. It seems that an initiator was the animal using its pec fin and that was the only behavior evaluated? Might want to indicate what the definition was for this. An operational definition of each behavior evaluated is needed as well. Analyses seemed appropriate.

Results & Discussion 

Generally, the organization was fine. A lot of statistics to wade through, although the authors attempted to keep the organization scheme clear. There were a few times where the discussion was lacking and needed some additional explanation/speculation and recommendations. There were lots of "data was" in this section. Please correct to "data were". Also, line 17 used ref #50, but there are 2 50s unfortunately. Be sure to correct all the references from here on down. The pec fin results need to clarify if the contact occurred during synchronous swims and whether those were active swims or rest swims. I am not sure we know what happens with pec fin contact between each type of swim really. . .Might be worth considering. Also, time of day should be considered a little more in terms of ecological history maybe. . .is there a reason that the different species might have different responses? Most likely related to housing and experience of animals, but might be worth addressing to some degree. Line 205 needs a new paragraph for "The frequency of socio-sex..." info. Too much for one paragraph and new DV. The use of "To resume, " for discussion portions is disorienting. Would recommend a different transition. Lines 216-219 - this discussion of time of day differences was difficult to follow. I am not exactly sure why the conclusion follows the beginning of this sentence.

Separation section needs more work. Right now, I am concerned that it could be used against captive facilities and temporary separations the way it is written. There is too much ambiguity in the discussion of what all of this means as well as what is meant by smaller social groups (might want to indicate in dyads if not solo). also, may want to discuss if separated were the animals in compatible pairs? Presumably yes, but this should probably be stated. Also, this discussion needs more development. Line 237 - social conditions is very ambiguous and should be clarified to ensure points are not misinterpreted by readers. Also, the inclusion of ref 9 to support the point should be used with caution in my opinion. The Waples and Gales paper had some issues with it that should be considered. 

Housing pool - Lines 255-256 - the comment about less space leads to more contact hypothesis. . .where would this hypothesis have come from? The way it is written makes the space sound like they can't get away from one another, I am not sure that is a fair interpretation. Might want to re-phrase this point somehow. Also, Lines 260-263 - another hypothesis without any real evidence for belief, and how is "easier inter-individual relationships" defined. Seems ambiguous as does the "conditions that do not allow individuals to engage in . . ." In my experience, some of the best bonding comes in very small, tight knit groups and areas. Although the space for fast swims and tight turns is a good one since those are harder to do with less space at the surface. Do social interactions such as these occur at depth or just surface?

Enrichment - fine in general. Would suggest some paragraphs as move through different species or points. Also, would like to see some discussion in Lines 309 - 333 about potential species difference and individual differences. Depending on the animals' training histories, trainer people may be very reinforcing and much preferred for attention rather than socializing with conspecifics. This is one topic that is not really addressed. Would suggest adding some discussion of the training history as a general construct. Lines 311-312 - often is used 2 times. Line 325 need to add "that" to the fact [that] BDs.…"

Disturbances - fine. the end of the discussion needs a little more clarification. I would recommend that authors tie up the last few sentences. There were several ideas presented and were somewhat disjointed. Why would sensitivity to noise, but noisy construction not disrupt play? THe last sentence is very confusing.

Public - results are fine. discussion needs a bit more. Lines 430-432 I think the conclusion provided about visual barriers is too much of a generalization from the terrestrial literature. This issue was not assessed in the current analyses so you can only speculate about the need for visual barriers when your whole premise is based on noise. I do not disagree with the conclusion, just that your data do not support it currently. Line 432 - don't end sentence with "they want to. . ." Think of a different way to say it and no preposition at the end.

Conclusions - helped to understand general findings. Think there could have been a little more discussion of certain points rather than reiterating all of the findings. Line 445 - do you think there was competition over the toys for the different species? or do you think species differences or individual experiences to different types of external stimuli affected the responses observed? Could humans be strong CSs and elicit certain responses from BDs that may not be present for the other two species? Vigilance might be a reasonable explanation but could there be something else? Would they prevent the interaction? 

line 460 - what is a high frequency of agonistic interaction? I would be careful with using relative terms like this without quantifying them, especially given today's world of cetacean welfare.

Lines 462-465 - did not follow this statement fully. might want to clarify point.

Some of references were not complete.

Author Response

Having read several other papers by the authors on similar topics, the authors did a good job of parsing out the data for this study to contribute additional information to the field. There is some redundancy between the measures reported in this paper and a couple of other publications, but I believe the data are new so acceptable. 

Yes, this data is part of the same dataset , but I was unable to publish all of it in a single paper. Which is why I spitted it.

The abstract was helpful and readable and accurate.

The intro covered the majority of the available literature. The authors did a nice job covering most of the cetacean literature as well as pulling in relevant terrestrial literature. The use of theses and dissertations was present, but I will leave that issue to the editors of the journal. Line 75, I would recommend adding in some way - "recently investigated but not published [40], and the modulation..." to address one of those unpublished sources. The sentence on socio-sexual behavior in line 77 is widowed and needs a home and probably a little more discussion. It was the only behavior not expanded on in the intro. Otherwise, all of the major DVS (pec fin, other contact, play, social play, agonistic intx, affiliative swims, socio-sex) were discussed within the current literature. 

I added more information about socio-sexual interactions.

The method was described well except for how the coding of the videos was actually conducted. Perhaps I missed this information? Also, was reliability assessed? All incidents were recorded for the key behaviors, so I am assuming frequency of bouts of the various social interactions were the used measures? What about the pec fin contact coding. . .I was a little surprised by the results in some cases. It seems that an initiator was the animal using its pec fin and that was the only behavior evaluated? Might want to indicate what the definition was for this. An operational definition of each behavior evaluated is needed as well. Analyses seemed appropriate.

L 155: Videos were visually analysed

I added details about the video analysis.

For this study, I did not conduct any reliability test because there was no ethologist in my lab and caretakers were not really cooperative for such additional work. However, I did inter-rater reliability tests together with experts on dolphins videos with high reliability scores in the past.

Yes, frequencies were used. I also recorded durations for bouts, but did not analyse it here, it would have been way too much results.

About pec contact, you are right, I only recorded the behaviour for the initiator (the pec contact giver). I did record who was giving a contact to who but I did not include this data here.

I added a table describing all analysed behaviours.

Results & Discussion 

Generally, the organization was fine. A lot of statistics to wade through, although the authors attempted to keep the organization scheme clear. There were a few times where the discussion was lacking and needed some additional explanation/speculation and recommendations. There were lots of "data was" in this section. Please correct to "data were". Also, line 17 used ref #50, but there are 2 50s unfortunately. Be sure to correct all the references from here on down. The pec fin results need to clarify if the contact occurred during synchronous swims and whether those were active swims or rest swims. I am not sure we know what happens with pec fin contact between each type of swim really. . .Might be worth considering. Also, time of day should be considered a little more in terms of ecological history maybe. . .is there a reason that the different species might have different responses? Most likely related to housing and experience of animals, but might be worth addressing to some degree. Line 205 needs a new paragraph for "The frequency of socio-sex..." info. Too much for one paragraph and new DV. The use of "To resume, " for discussion portions is disorienting. Would recommend a different transition. Lines 216-219 - this discussion of time of day differences was difficult to follow. I am not exactly sure why the conclusion follows the beginning of this sentence.

I corrected the "data was"

I corrected the references order.

I included all pectoral fin contacts in the category. Animals could be swimming, resting or interacting when it occurred.

I discussed the need to go deeper into the different kinds of pec contacts.

I deleted the "to resume"

I modified the discussion part about time of the day.

Separation section needs more work. Right now, I am concerned that it could be used against captive facilities and temporary separations the way it is written. There is too much ambiguity in the discussion of what all of this means as well as what is meant by smaller social groups (might want to indicate in dyads if not solo). also, may want to discuss if separated were the animals in compatible pairs? Presumably yes, but this should probably be stated. Also, this discussion needs more development. Line 237 - social conditions is very ambiguous and should be clarified to ensure points are not misinterpreted by readers. Also, the inclusion of ref 9 to support the point should be used with caution in my opinion. The Waples and Gales paper had some issues with it that should be considered. 

Right, I discussed it more. The thing is that, here, in this context, separation was rarely needed and it was quite easy to notice the impact on the animals' behaviour. But I did not mean to generalize and say that separation is always negative.

Housing pool - Lines 255-256 - the comment about less space leads to more contact hypothesis. . .where would this hypothesis have come from? The way it is written makes the space sound like they can't get away from one another, I am not sure that is a fair interpretation. Might want to re-phrase this point somehow. Also, Lines 260-263 - another hypothesis without any real evidence for belief, and how is "easier inter-individual relationships" defined. Seems ambiguous as does the "conditions that do not allow individuals to engage in . . ." In my experience, some of the best bonding comes in very small, tight knit groups and areas. Although the space for fast swims and tight turns is a good one since those are harder to do with less space at the surface. Do social interactions such as these occur at depth or just surface?

Actually it was here... the small pool was really really small. Here, I just try to interpret my own results, I have less body contacts in the small pool. But this of course, might only be true for this group. The small pool was really small and animals were really energetic (I saw many groups of bottlenose, but those that Chinese parks own are particularly energetic, and the experts who visited me can confirm this). It is hard for me to emphasize the issues in Chinese parks in such a paper but it surely played a huge role in my data and results would surely not be the same in European groups of dolphins.

Enrichment - fine in general. Would suggest some paragraphs as move through different species or points. Also, would like to see some discussion in Lines 309 - 333 about potential species difference and individual differences. Depending on the animals' training histories, trainer people may be very reinforcing and much preferred for attention rather than socializing with conspecifics. This is one topic that is not really addressed. Would suggest adding some discussion of the training history as a general construct. Lines 311-312 - often is used 2 times. Line 325 need to add "that" to the fact [that] BDs.…"

Disturbances - fine. the end of the discussion needs a little more clarification. I would recommend that authors tie up the last few sentences. There were several ideas presented and were somewhat disjointed. Why would sensitivity to noise, but noisy construction not disrupt play? THe last sentence is very confusing.

Public - results are fine. discussion needs a bit more. Lines 430-432 I think the conclusion provided about visual barriers is too much of a generalization from the terrestrial literature. This issue was not assessed in the current analyses so you can only speculate about the need for visual barriers when your whole premise is based on noise. I do not disagree with the conclusion, just that your data do not support it currently. Line 432 - don't end sentence with "they want to. . ." Think of a different way to say it and no preposition at the end.

Conclusions - helped to understand general findings. Think there could have been a little more discussion of certain points rather than reiterating all of the findings. Line 445 - do you think there was competition over the toys for the different species? or do you think species differences or individual experiences to different types of external stimuli affected the responses observed? Could humans be strong CSs and elicit certain responses from BDs that may not be present for the other two species? Vigilance might be a reasonable explanation but could there be something else? Would they prevent the interaction? 

line 460 - what is a high frequency of agonistic interaction? I would be careful with using relative terms like this without quantifying them, especially given today's world of cetacean welfare.

Lines 462-465 - did not follow this statement fully. might want to clarify point.

Some of references were not complete.

Reviewer 2 Report

Key issues:

This is an interesting and valuable study but lacks a central narrative and conclusions. I appreciate the amount of data that was collected and analysed, but it should be presented more clearly for the reader to follow, making key points throughout the results/discussion section. In some cases, the result/discussion section lacked clarity with respect to why the analysis was undertaken and would be important with respect to welfare management (e.g. stress why laterality of fin contact matters).

The first sentence in the 'Simple Summary' reads as: 'Few welfare indicators have been validated for odontocete species'. This study does not validate welfare indicators (welfare was only measured using video observation as a proxy, there was no attempt to verify this measurement), it applies the ones commonly used in bottlenose dolphins to two other captive odontocetes. While this is stressed as a novelty of the study, there is little comparison or conclusion regarding this proxy as a measure of welfare in the other two species found in the conclusion, despite abstract and simple summary suggesting so.

Other points:

Simple summary:

  • The simple summary is somewhat confusing to read and conclusions are not clear, please restructure and highlight the importance of the study. (This should be stressed in the first few sentences). The same issue applies for the Abstract: e.g. why is this study novel and important (e.g.what are the species-specific management approaches suggested)?
  • Line 14: diurnal patterns for most of the social behaviours analyzed - only diurnal patterns were analysed? This suggests they did not experience nocturnal patterns?

Abstract

  • Define 'a side bias for pectoral contacts' and the importance of in a behavioural sense. Why is pectoral contact important?

Introduction:

  • Line 41: n increasingly popular research subject in recent years - popular because maintaining welfare is important. Would rephrase this as it sounds like welfare considerations are merely a 'trend'
  • Line 55: important points made, but needs to be re-written (currently too colloquial)
  • Line 77: lacks dot to end sentence, the sentence seems out of place, please revise
  • Please highlight/stress why this research is of importance with respect to animal welfare in the final paragraph of introduction

Methods:

  • Data collection: please clarify how many hours of data were collected: it is unclear (45 minutes per week split into three sessions of 15 minutes?)
  • Line 140: 'cameras with a fair enough quality to' - consider rewriting too colloquial
  • Line 145: all animals together: “not separated”, group divided in subgroups: 146 “separated”) - is a distinction made between sub group sizes?
  • Line 147: presence of visitors (“none”, “few” or “many”) - specify the difference between few and too many?
  • Line 148: and every unusual event that occurred (“disturbance”) - provide an example?
  • Were cetacean specific measures (sex, age etc.) included in the analysis?

Results/ Discussion

  • As above, does a session refer to one 15min recording, please clarify
  • 3.1. Pectoral Fin Contact Laterality: stress the importance of these findings
  • Line 182: Time of the day - while interesting, the authors fail to stress why these findings are important from a welfare or management perspective.
  • Line 227-228: grammar 'altogether' is not the right word choice
  • Line 235: This lower level of playfulness when separated has already been shown in these  groups when analyzing solitary play ... - these results are interesting and potentially important from a welfare perspective, please discuss them further 
  • Line 271: In addition, when giving polar bears (Ursus 272 maritimus) the opportunity of accessing their indoor enclosures, they engaged more often in social play ([56]). - different species (not a marine mammal), please find more relevant example here

Conclusion:

  • The conclusion needs to be re-written to highlight the most important findings only, contrast those across the three species and then suggest management approaches. 

Author Response

This is an interesting and valuable study but lacks a central narrative and conclusions. I appreciate the amount of data that was collected and analysed, but it should be presented more clearly for the reader to follow, making key points throughout the results/discussion section. In some cases, the result/discussion section lacked clarity with respect to why the analysis was undertaken and would be important with respect to welfare management (e.g. stress why laterality of fin contact matters).

The first sentence in the 'Simple Summary' reads as: 'Few welfare indicators have been validated for odontocete species'. This study does not validate welfare indicators (welfare was only measured using video observation as a proxy, there was no attempt to verify this measurement), it applies the ones commonly used in bottlenose dolphins to two other captive odontocetes. While this is stressed as a novelty of the study, there is little comparison or conclusion regarding this proxy as a measure of welfare in the other two species found in the conclusion, despite abstract and simple summary suggesting so.

This is a preliminary investigation. It does not aim to validate indicators but to determine those that can be interesting to further study.

Other points:

Simple summary:

  • The simple summary is somewhat confusing to read and conclusions are not clear, please restructure and highlight the importance of the study. (This should be stressed in the first few sentences). The same issue applies for the Abstract: e.g. why is this study novel and important (e.g.what are the species-specific management approaches suggested)?
  • Modified
  • Line 14: diurnal patterns for most of the social behaviours analyzed - only diurnal patterns were analysed? This suggests they did not experience nocturnal patterns?
  • I did not record at night

Abstract

  • Define 'a side bias for pectoral contacts' and the importance of in a behavioural sense. Why is pectoral contact important?
  • I do not have the space to detail it in the abstract... I discussed it more in the discussion part. Should I add a part to specifically discuss the influence of the context on the side animals used? I am worried it would be too much information.

Introduction:

  • Line 41: n increasingly popular research subject in recent years - popular because maintaining welfare is important. Would rephrase this as it sounds like welfare considerations are merely a 'trend'
  • I rephrased it
  • Line 55: important points made, but needs to be re-written (currently too colloquial)
  • I rewrote it
  • Line 77: lacks dot to end sentence, the sentence seems out of place, please revise
  • My mistake, I corrected it.
  • Please highlight/stress why this research is of importance with respect to animal welfare in the final paragraph of introduction
  • I did

Methods:

  • Data collection: please clarify how many hours of data were collected: it is unclear (45 minutes per week split into three sessions of 15 minutes?)
  • Clarified
  • Line 140: 'cameras with a fair enough quality to' - consider rewriting too colloquial
  • Rephrased
  • Line 145: all animals together: “not separated”, group divided in subgroups: 146 “separated”) - is a distinction made between sub group sizes?
  • No, groups were always small when separated (1-3 animals)
  • Line 147: presence of visitors (“none”, “few” or “many”) - specify the difference between few and too many?
  • I added a table
  • Line 148: and every unusual event that occurred (“disturbance”) - provide an example?
  • I added a table
  • Were cetacean specific measures (sex, age etc.) included in the analysis?
  • Since our sample was too small to investigate such features, we did not include it. However, we included the individual ID as a random factor in each model to account for inter individual differences

Results/ Discussion

  • As above, does a session refer to one 15min recording, please clarify
  • Clarified
  • 3.1. Pectoral Fin Contact Laterality: stress the importance of these findings
  • I did
  • Line 182: Time of the day - while interesting, the authors fail to stress why these findings are important from a welfare or management perspective.
  • I discussed it further
  • Line 227-228: grammar 'altogether' is not the right word choice
  • Corrected
  • Line 235: This lower level of playfulness when separated has already been shown in these  groups when analyzing solitary play ... - these results are interesting and potentially important from a welfare perspective, please discuss them further 
  • I did discuss it further
  • Line 271: In addition, when giving polar bears (Ursus 272 maritimus) the opportunity of accessing their indoor enclosures, they engaged more often in social play ([56]). - different species (not a marine mammal), please find more relevant example here
  • I did not find an example in a closer species

Conclusion:

  • The conclusion needs to be re-written to highlight the most important findings only, contrast those across the three species and then suggest management approaches. 
  • Done

Reviewer 3 Report

Dear Serres et al. I applaud you for the amount of work conducted, collecting what appears to be a considerable amount of data. There is a great need for quantitative studies on cetacean welfare in captivity and comparisons among species are incredibly useful, however, while I support your research topic, the manuscript contains some fundamental flows in reasoning and it should be rewritten with the help of a Native english speaker. Currently, the manuscript is very hard to comprehend, contains too many grammatical mistakes and lacks several definitions. Moreover, I'm not convinced that GLMM have been used in the correct way and I suggest to use simpler statistics or provide enough information on model selection and usage.

The short summary and the introduction should be rewritten almost entirely and it would be beneficial to separate results from discussion. Following, I provide some more detailed comments on each section and a line by line review.

Title: environmental parameters  might be misleading since it can be read as the natural environment and characteristics such as temperature, salinity, etc. Instead, the study examines housing facilities, about habitat characteristics. Also, impacted has a negative consequence while in this case I would use influence.

Simple Summary: It miss a brief description of the methodology so the reader is left unsure on where results comes from (line 13 to 18).

Abstract: the study suggests to use social interactions and pectoral contacts as welfare indicators but without providing a sound rational for it. The abstract (and throughout the text) also mention differences in individual responses but these are not tested in the study (or at least it is not shown in the manuscript).

Introduction: the introduction would benefit from a proper discussion on the various ways to define welfare with a mention to the five freedoms, generally presented as an indication of welfare. The introduction presents several grammar mistakes, repetitions and should be rewritten to improve English and conceptual flow. It currently presents welfare indicators in odontocetes then describe social behaviours in odontocetes to then introduce welfare in other species and then social behaviour in dolphins.

Methods: housing conditions and group compositions are well described but the fate of two out of three calves isn’t mentioned, nor the potential effect on females behaviour. Also it lacks a clear definition of behaviours (i.e. socio-sexual interactions, other social behaviour, etc.) and predictor variables (disturbance, enrichment, etc.). A table with clear definitions of the behaviours and predictors analysed would be highly beneficial. The manuscript doesn’t provide any description of model selection method nor state whether predictors were analysed separately or all together. There is no mention whether interactions among predictors have been analysed. A list of the model used should be provided. Also, the sampling period (each 15min video) should be used as a random effect since it represent a nested design. I suggest provide further justification for using GLMM or using a simpler statistical analyses since the author is currently simply comparing group averages.

Results and Discussion: the manuscript would greatly benefit from separating results and discussion in two sections. Please provide a table with model selection (if relevant) and one table with the parameters coefficients (including-value, p-value and weight if relevant).

Conclusions: the manuscript infer conclusions that are not supported by the data

Throughout the paper, in text citation should not be within round brackets.

Line by line review:

11-12: “Since these species odontocetes are often highly social animals, social behaviors are interesting ones could be used to investigate…”

13: the use of diurnal pattern throughout the text is incorrect. Diurnal pattern is meant as something happening in the day and not at night but here the author seems to refer to a difference in behaviours depending on the time of day. I suggest change diurnal pattern with “time of day” throughout the entire manuscript.

13: “accessibility of to several pools”

17: I agree that agonistic interactions, body contact and social play are influenced by the habitat characteristics (not environmental!) but it is too big of a leap to say that they can be used to assess welfare. At present, the manuscript lack a clear link between these behaviours and welfare.

18: The author didn’t test for individual differences or these are not presented within the manuscript so it is inappropriate to mention them here. The author also specify this in line 179, thus contradicting herself.

23: “investigated the pectoral contact laterality” and “routine environmental conditions habitat characteristics”

24: unclear what latin name belong to who, keep consistency by using common name first, then abbreviation, then latin name)

32-33 too big of a leap, see previous comment to line 17

34: see previous comment to line 18

41: provide reference

42: the term “positive emotions” is redundant since in the same line are mentioned positive and negative emotional state

45: remove “welfare indicators”, is redundant; remove “Those Welfare indicators…”

48-49 should be moved to line 51

53 on other species that might have different social lives and might not express their emotional state as bottlenose

55-59 the entire section is a repetition

64-65 repetition. also, I disagree that perctoral contact decrease the frequency of aggressive behaviours. Pectoral contacts are generally used in different behavioural context but are not the cause of decrease aggressive behaviours!

67: “which are potentially interesting for assessing can be used to assess welfare too”

74: “The behavioural context of the occurrence of such body contacts in which such body contacts occur“

77: Please provide definition for socio-sexual interactions. Also the sentence seems to finish mid-air and it is out of context

84-86: repetition

87: “factors that might impact individuals’ welfare on the social behaviour”

97: Please provide definition of subgroups

103: what happened to the two calves who we present for only 2 weeks? Were they separated from their mother? If so, how this impacted the mothers’ behaviour? Was that taken into consideration in the analyses?

114: How can the dolphin be fed only 3.5kg a day? Does this represent the whole daily requirements?

120: Please provide a definition for enrichment

126: Why building a common ethogram for the three species?

147: define what “few” and “many” mean (aka how many dolphins were “few”?)

151: The author mention previously defined social behaviour but didn’t actually provide a definition of these

152: 47 reference leads to two different publications in the bibliography

160: Each 15 min video should be added as a random factor in the GLMM as they represent a nested study design. Also, it is incorrect to say that the frequency of individual ID is used as random factor. Frequency has nothing to do with the random factor ID. I suggest revise the use of GLMM to familiarize better with this technique or use a simpler one.

162: Please provide reference for using VIF<10 as no collinearity, Zuur et al. 2010 in Methods in Ecology and Evolution actually suggest using 3 or less.

163-164: Please provide further details

166: Strongly suggest to separate results from discussion

171: Why age categories were not included in the models as predictors? Juveniles and calves tend to be more playful and display different social behaviours than adults. Also previous studies suggest that mum and calf pairs present visual laterality that would necessarily influence pectoral contacts laterality too

176: Please provide reference for the study by citing the name of the authors

177 “analysed pectoral fin contact bias in males, females and overall population suggested an ambidextrousness

198: Please provide definition for agonistic interactions

227: when analysing frequency of pectoral contact with individuals altogether vs separate, are the total number of individuals taken into account? For instance, if I count how many pectoral contact I see during 15min intervals, I will have a higher probability of counting more contacts if I have 5 dolphins instead of 2 just because of the total number of dolphins present.

237: Please define social swimming

239: Was sex taken into account in looking at difference in antagonistic and socio-sexual behaviour? Male individuals will have different types and rate of antagonistic and sexual behaviours with other male compared to females

243: Are pectoral rubbing and pectoral contact the same? If so, choose one and be consistent, otherwise please provide more information

258: until now the author treated pectoral contacts and socio-sexual behaviours as different, now they appear to be the same. Please specify and be consistent with the definition of each behaviour.

262-266: very hard to understand, should be rewritten entirely

269: provide reference for this statement

271: Lack clear connection with welfare

274: The paragraph needs a concluding statement to better place the study’s results into context

277: Please provide an explanation why humans and fish are considered enrichment. They could be equally considered as disturbance and food, respectively

311: define it. Missing verb between EAFPs and often

312-327: this section should be completely rewritten, there are too many missing verbs, and grammatical errors to comment on them all

333-337: well done

341: it lacks a concluding statement; what would the author suggest in regards to captive cetacean?

346: define social events

347: define noisy events

348: define other disturbances

379-380: I feel like behavioural expression of emotions might bit too humanising too much since the author did not tested for it. Moreover, the author makes the incorrect assumption that all disturbances would affect the dolphins in the same way.

383-386: a higher frequency of fast swimming might also reflect a defensive response and time budget would be a consequence not a cause of decrease antagonistic behaviour

388-391: the sentence is very hard to comprehend I suggest rewrite it

419: what are the negative behavioural indicators? Also the example on Rhino provided in the following sentence is inappropriate since it mention physiological responses and not behavioural ones!

428-432: the sentence is too long and is very hard to comprehend I suggest rewrite it

443: “the presence of toys or of human”

454: the author didn’t examined or did not commented on differences across individuals so it is inappropriate to mention them here

460: define social problems

Author Response

Dear Serres et al. I applaud you for the amount of work conducted, collecting what appears to be a considerable amount of data. There is a great need for quantitative studies on cetacean welfare in captivity and comparisons among species are incredibly useful, however, while I support your research topic, the manuscript contains some fundamental flows in reasoning and it should be rewritten with the help of a Native english speaker. Currently, the manuscript is very hard to comprehend, contains too many grammatical mistakes and lacks several definitions. Moreover, I'm not convinced that GLMM have been used in the correct way and I suggest to use simpler statistics or provide enough information on model selection and usage.

The short summary and the introduction should be rewritten almost entirely and it would be beneficial to separate results from discussion. Following, I provide some more detailed comments on each section and a line by line review.

Title: environmental parameters  might be misleading since it can be read as the natural environment and characteristics such as temperature, salinity, etc. Instead, the study examines housing facilities, about habitat characteristics. Also, impacted has a negative consequence while in this case I would use influence.

I modified the title

Simple Summary: It miss a brief description of the methodology so the reader is left unsure on where results comes from (line 13 to 18).

I added it

Abstract: the study suggests to use social interactions and pectoral contacts as welfare indicators but without providing a sound rational for it. The abstract (and throughout the text) also mention differences in individual responses but these are not tested in the study (or at least it is not shown in the manuscript).

I think I should still mention it because they exist. It would have been a way too long manuscript that no journal would have accepted if I had gone to the individual level.

Introduction: the introduction would benefit from a proper discussion on the various ways to define welfare with a mention to the five freedoms, generally presented as an indication of welfare. The introduction presents several grammar mistakes, repetitions and should be rewritten to improve English and conceptual flow. It currently presents welfare indicators in odontocetes then describe social behaviours in odontocetes to then introduce welfare in other species and then social behaviour in dolphins.

I reorganized and added more definitions of welfare.

Methods: housing conditions and group compositions are well described but the fate of two out of three calves isn’t mentioned, nor the potential effect on females behaviour. Also it lacks a clear definition of behaviours (i.e. socio-sexual interactions, other social behaviour, etc.) and predictor variables (disturbance, enrichment, etc.). A table with clear definitions of the behaviours and predictors analysed would be highly beneficial. The manuscript doesn’t provide any description of model selection method nor state whether predictors were analysed separately or all together. There is no mention whether interactions among predictors have been analysed. A list of the model used should be provided. Also, the sampling period (each 15min video) should be used as a random effect since it represent a nested design. I suggest provide further justification for using GLMM or using a simpler statistical analyses since the author is currently simply comparing group averages.

I added the information. Both calves died.

I added a table to describe each behaviour and each predictor.

I added precisions about the GLMMs.

As you mentioned, I did repeated measurements, and I can therefore not use many of these simpler stat analysis. 

Results and Discussion: the manuscript would greatly benefit from separating results and discussion in two sections. Please provide a table with model selection (if relevant) and one table with the parameters coefficients (including-value, p-value and weight if relevant).

I included these values in the text. A table showing each model's selection would be way too long.

Conclusions: the manuscript infer conclusions that are not supported by the data

I modified the conclusion

Throughout the paper, in text citation should not be within round brackets.

Corrected

Line by line review:

11-12: “Since these species odontocetes are often highly social animals, social behaviors are interesting ones could be used to investigate…”

Done

13: the use of diurnal pattern throughout the text is incorrect. Diurnal pattern is meant as something happening in the day and not at night but here the author seems to refer to a difference in behaviours depending on the time of day. I suggest change diurnal pattern with “time of day” throughout the entire manuscript.

Done

13: “accessibility of to several pools”

Corrected

17: I agree that agonistic interactions, body contact and social play are influenced by the habitat characteristics (not environmental!) but it is too big of a leap to say that they can be used to assess welfare. At present, the manuscript lack a clear link between these behaviours and welfare.

Modified

18: The author didn’t test for individual differences or these are not presented within the manuscript so it is inappropriate to mention them here. The author also specify this in line 179, thus contradicting herself.

Removed

23: “investigated the pectoral contact laterality” and “routine environmental conditions habitat characteristics”

Done

24: unclear what latin name belong to who, keep consistency by using common name first, then abbreviation, then latin name)

Done

32-33 too big of a leap, see previous comment to line 17

Corrected

34: see previous comment to line 18

Corrected

41: provide reference

Added

42: the term “positive emotions” is redundant since in the same line are mentioned positive and negative emotional state

Corrected

45: remove “welfare indicators”, is redundant; remove “Those Welfare indicators…”

Modified

48-49 should be moved to line 51

I modified this part

53 on other species that might have different social lives and might not express their emotional state as bottlenose

Modified

55-59 the entire section is a repetition

Modified

64-65 repetition. also, I disagree that perctoral contact decrease the frequency of aggressive behaviours. Pectoral contacts are generally used in different behavioural context but are not the cause of decrease aggressive behaviours!

Rephrased

67: “which are potentially interesting for assessing can be used to assess welfare too”

Modified

74: “The behavioural context of the occurrence of such body contacts in which such body contacts occur“

Corrected

77: Please provide definition for socio-sexual interactions. Also the sentence seems to finish mid-air and it is out of context

Added and modified

84-86: repetition

Modified

87: “factors that might impact individuals’ welfare on the social behaviour”

Corrected

97: Please provide definition of subgroups

Modified

103: what happened to the two calves who we present for only 2 weeks? Were they separated from their mother? If so, how this impacted the mothers’ behaviour? Was that taken into consideration in the analyses?

I added the information. The two mothers did not take care of their calves who both died. This was included in the analysis as an unusual social event.

114: How can the dolphin be fed only 3.5kg a day? Does this represent the whole daily requirements?

Finless porpoises are not dolphins, but porpoises. They are way smaller. Bottlenose dolphins were fed about 10-13kg a day.

120: Please provide a definition for enrichment

Done

126: Why building a common ethogram for the three species?

The goal was to compare their responses, I wanted to use as many common behaviours as I could. I still added social play for BDs even if I almost never observed it in FPs.

147: define what “few” and “many” mean (aka how many dolphins were “few”?)

Few and many are adjectives given to the public density, not to the dolphins... I added a table to clarify this.

151: The author mention previously defined social behaviour but didn’t actually provide a definition of these

I added a table

152: 47 reference leads to two different publications in the bibliography

Corrected

160: Each 15 min video should be added as a random factor in the GLMM as they represent a nested study design. Also, it is incorrect to say that the frequency of individual ID is used as random factor. Frequency has nothing to do with the random factor ID. I suggest revise the use of GLMM to familiarize better with this technique or use a simpler one.

It was a typo mistake. I am quite familiar with this technique and after consulting several stat experts, the analysis I did is the most suitable one.

162: Please provide reference for using VIF<10 as no collinearity, Zuur et al. 2010 in Methods in Ecology and Evolution actually suggest using 3 or less.

Added. The literature says both. Over three should raise concern and over 10 should be corrected. I had all my VIF values between 1 and 2.

163-164: Please provide further details

Done

166: Strongly suggest to separate results from discussion

Done

171: Why age categories were not included in the models as predictors? Juveniles and calves tend to be more playful and display different social behaviours than adults. Also previous studies suggest that mum and calf pairs present visual laterality that would necessarily influence pectoral contacts laterality too

Calves were not included in the analysis. Only adults were. One calf was present for a few months, but his behaviour was not included in the study.

176: Please provide reference for the study by citing the name of the authors

Corrected

177 “analysed pectoral fin contact bias in males, females and overall population suggested an ambidextrousness

Corrected

198: Please provide definition for agonistic interactions

I added a table

227: when analysing frequency of pectoral contact with individuals altogether vs separate, are the total number of individuals taken into account? For instance, if I count how many pectoral contact I see during 15min intervals, I will have a higher probability of counting more contacts if I have 5 dolphins instead of 2 just because of the total number of dolphins present.

I weighted the number of each behaviour by the number of animals present before running the analysis. Added in Mat Meth.

237: Please define social swimming

Done

239: Was sex taken into account in looking at difference in antagonistic and socio-sexual behaviour? Male individuals will have different types and rate of antagonistic and sexual behaviours with other male compared to females

Right, but no, it was not taken into account. Once again, the results would have been way too long if I had included a sex analysis for each model. And this was not the goal here to go into inter-sex differences. If you still think it would be better adding it, I can re run the analysis with a sex predictor in each model.

243: Are pectoral rubbing and pectoral contact the same? If so, choose one and be consistent, otherwise please provide more information

My mistake. Corrected

258: until now the author treated pectoral contacts and socio-sexual behaviours as different, now they appear to be the same. Please specify and be consistent with the definition of each behaviour.

I added a table

262-266: very hard to understand, should be rewritten entirely

I rewrote it

269: provide reference for this statement

I do not have any reference for this, this is a personal observation.

271: Lack clear connection with welfare

Re-organised

274: The paragraph needs a concluding statement to better place the study’s results into context

Re-organised

277: Please provide an explanation why humans and fish are considered enrichment. They could be equally considered as disturbance and food, respectively

Right. We decided that a stimuli would go in enrichment or in disturbance depending on the purpose of this stimuli. If it was provided to be enriching, we labelled it as enrichment, if not, as disturbance. But, you are right, we could have misjudged these stimuli, that's why we analysed the effects of each type of enrichment/disturbance in detail. The presence of divers could also become a kind of enrichment for animals, but we included it as a disturbance. We could have included all of them in a larger category instead of having two. If you think it would be better, we can re organise this.

311: define it. Missing verb between EAFPs and often

The verb is right after. "EAFPs often observed"

312-327: this section should be completely rewritten, there are too many missing verbs, and grammatical errors to comment on them all

I rewrote it

333-337: well done

341: it lacks a concluding statement; what would the author suggest in regards to captive cetacean?

I added discussion

346: define social events

I added a table

347: define noisy events

I added a table

348: define other disturbances

I added a table

379-380: I feel like behavioural expression of emotions might bit too humanising too much since the author did not tested for it. Moreover, the author makes the incorrect assumption that all disturbances would affect the dolphins in the same way.

Corrected

383-386: a higher frequency of fast swimming might also reflect a defensive response and time budget would be a consequence not a cause of decrease antagonistic behaviour

Rephrased

388-391: the sentence is very hard to comprehend I suggest rewrite it

I rewrote it

419: what are the negative behavioural indicators? Also the example on Rhino provided in the following sentence is inappropriate since it mention physiological responses and not behavioural ones!

I replaced this example and mentioned negative indicators.

428-432: the sentence is too long and is very hard to comprehend I suggest rewrite it

I rewrote it

443: “the presence of toys or of human”

Corrected

454: the author didn’t examined or did not commented on differences across individuals so it is inappropriate to mention them here

Corrected

460: define social problems

Done